# New insights into the ~74 ka Toba eruption from sulfur isotopes of polar ice cores

Laura Crick[1], Andrea Burke[1], William Hutchison[1], Mika Kohno[2], Kathryn A. Moore[3], Joel Savarino[4], Emily A. Doyle[5], Sue Mahony[6], Sepp Kipfstuhl[7], James W. B. Rae[1], Robert C. J. Steele[1], R. Stephen J. Sparks[6] and Eric W. Wolff[5]

[1]School Of Earth And Environmental Sciences, University Of St Andrews, United Kingdom
[2]Geoscience Center (GZG), Department of Geochemistry, Georg August Universität Göttingen, Germany
[3]Department Of Atmospheric Science, Colorado State University, USA
[4]Institut Des Géosciences De L'environnement, Grenoble, France
[5]Department Of Earth Sciences, University Of Cambridge, United Kingdom
[6]School Of Earth Sciences, University Of Bristol, United Kingdom
[7]Alfred-Wegener-Institute, Bremerhaven, Germany

**Correspondence**: Laura Crick (lc258@st-andrews.ac.uk)

**Abstract.** The ~74 ka Toba eruption was one of the largest volcanic events of the Quaternary. There is much interest in determining the impact of such a large event, particularly on the climate and hominid populations at the time. Although the Toba eruption has been identified in both land and marine archives as the Youngest Toba Tuff, its precise place in the ice core record is ambiguous. Several volcanic sulfate signals have been identified in both Antarctic and Greenland ice cores and span the Toba eruption $^{40}Ar/^{39}Ar$ age uncertainty. Here, we measure sulfur isotope compositions in Antarctic ice samples from the EDC and EDML ice cores at high temporal resolution across 11 of these potential Toba sulfate peaks to identify candidates with sulfur mass independent fractionation (S-MIF), indicative of an eruption whose plume reached altitudes at or above the stratospheric ozone layer. Using this method, we identify several candidate sulfate peaks that contain stratospheric sulfur. We further narrow down potential candidates based on the isotope signatures by identifying sulfate peaks that are due to a volcanic event at tropical latitudes. In one of these sulfate peaks at 73.67 ka, we find the largest ever reported magnitude of S-MIF in volcanic sulfate in polar ice, with a $\Delta^{33}S$ value of -4.75 ‰. As there is a positive correlation between the magnitude of the S-MIF signal recorded in ice cores and eruptive plume height, this could be a likely candidate for the Toba supereruption, with a plume top height in excess of 45 km. These results support the 73.7 ± 0.3 ka (1σ) ka $^{40}Ar/^{39}Ar$ age estimate for the eruption, with ice core ages of our candidates with the largest magnitude S-MIF at 73.67 and 73.74 ka. Finally, since these candidate eruptions occurred on the transition into Greenland Stadial 20, the relative timing suggests that Toba was not the trigger for the large Northern Hemisphere cooling at this time although we cannot rule out an amplifying effect.

# 1 Introduction

The Toba caldera is located in northern Sumatra (Indonesia) and contains the largest volcanic lake on Earth. The 100 x 30 km caldera marks the location of a supereruption which occurred around 74 ka and covered most of North Sumatra in ignimbrite (Chesner, 2012). Evidence for this eruption comes in the form of ash in marine cores from the Bay of Bengal, Indian Ocean, and South China Sea. On land, layers of ash in India, Malaysia, and as far as Lake Malawi in East Africa have been identified as the Youngest Toba Tuff (YTT) (Williams, 2012; Petraglia et al., 2012; Lane et al., 2013). In total, an area of ~40 million $km^2$ is estimated to have been covered in >5 mm of ash (Costa et al., 2016). The Toba eruption is estimated to be two orders of magnitude greater than the 1815 Tambora eruption (Chesner et al., 1991) making it of one of the largest eruptions of the Quaternary. Given the scale of the Toba eruption, there is great interest in determining the climate forcing associated with the Toba event and its timing for use as a stratigraphic marker (Oppenheimer, 2002).

Understanding the amount and height of sulfur ejected during a volcanic event is key to establishing the climatic impacts of an eruption. Volcanic eruptions release $SO_2$ and $H_2S$ into the atmosphere which is rapidly oxidised to sulfate. Sulfate aerosols scatter incoming solar radiation leading to cooling of the Earth's surface (Cole-Dai, 2010). In the troposphere, sulfur compounds are oxidised and precipitated out in days to weeks (Shaheen et al., 2013), whereas sulfate aerosols can remain in the stratosphere for months to years, leading to extensive global cooling (Toohey et al., 2019; Robock, 2000). For the Toba eruption, some studies have linked this event to a bottleneck in human evolution (Rampino and Ambrose, 2000) while others suggest that human evolution was mostly unaffected by the supereruption (Clarkson et al., 2020; Lane et al., 2013; Smith et al., 2018). Key to unravelling its potential for climatic forcing is determining the stratospheric sulfate loading associated with this event and its precise timing.

There are currently a number of estimates for the amount of sulfur erupted into the atmosphere during the Toba event and a large range have been used to model the potential climatic impact of Toba (Yost et al., 2018). A study using melt inclusions in the YTT estimate the sulfur aerosol load to be 33 Tg S (Chesner and Luhr, 2010), slightly higher than the 1815 Tambora eruption (27 – 29 Tg S, (Self et al., 2004)). Subsequently, Costa et al., (2014) used the same melt inclusion data, with a different set of initial assumptions, and new estimates of the volume of the eruption to re-estimate the sulfur loading of Toba to be 850–1750 Tg S. These differences in estimates of aerosol load highlight how differences in assumptions can lead to large uncertainties associated with this technique. Furthermore, these petrological estimates can only constrain total sulfur emitted, with no distinction on how much reached the stratosphere which is crucial for understanding the potential climatic impact.

An alternative means of estimating sulfur emissions from past volcanic eruptions is through analysis of sulfate from polar ice cores. Sulfate concentration measurements are available from both hemispheres with records extending back to 110 ka for

Greenland (Mayewski et al., 1997) and 800 ka for Antarctica (Wolff et al., 2010a). Peaks in sulfate concentration above background concentrations can be identified as volcanic events. Large, low-latitude eruptions can distribute sulfate globally and give rise to a coincident bipolar peak in sulfate (Sigl et al., 2013; Gao et al., 2007; Zielinski, 2000). Zielinski et al., (1996) identified the largest sulfate peak around the time of Toba in the GISP2 Greenland ice core, T1 in this study, and used this to estimate a sulfur loading of 230–1140 Tg S over ~6 years. This range lies between the two melt inclusion estimates

described above and overlaps the upper estimate.

A further benefit of using ice core estimates of sulfur loading is that the isotopic composition of the volcanic sulfate deposited in ice cores can be used to determine if the sulfate was injected into the stratosphere to altitudes at or above the ozone layer (e.g. Savarino et al., 2003; Baroni et al., 2007; Gautier et al., 2019). Most chemical and physical processes

fractionate isotopes as a function of their masses. A notable exception is when sulfur dioxide is exposed to UV radiation it acquires a sulfur mass independent fractionation (S-MIF) signature in the form of non-zero $\Delta^{33}S$ values (Farquhar et al., 2001) here:

$$\Delta^{33}S = \delta^{33}S - (\delta^{34}S+1)^{0.515} +1 \tag{1}$$

where 0.515 is the equilibrium fractionation factor given by the ratio of the differences in the reduced masses and $\delta^{x}S$ is

$$\delta^{x}S= \frac{(^{x}S/^{32}S)_{sample}}{(^{x}S/^{32}S)_{reference}} -1 \tag{2}$$

and x = 33 or 34 with Vienna-Canyon Diablo Troilite (V-CDT) as the reference.

In the modern atmosphere, the ozone layer prevents the relevant wavelengths in the UV spectrum from reaching the Earth's surface. Therefore any volcanic $SO_2$ erupted below the ozone layer will fall along the terrestrial mass dependent

fractionation line, but an eruption into or above the ozone layer will inherit a S-MIF signal of non-zero $\Delta^{33}S$ (Savarino et al., 2003). As the ozone layer is in the lower stratosphere, the presence of S-MIF in volcanic sulfate can identify stratospheric sulfate. Sulfur isotope compositions in volcanic sulfate from snow pits and ice cores have been measured on many of the major eruptions within the last 2600 years (e.g. Savarino et al., 2003; Baroni et al., 2007; Cole-Dai et al., 2009; Gautier et al., 2019).  This isotope signal varies across the duration of an eruption with a number of these studies having shown that the

$\Delta^{33}S$ evolves from positive to negative values (e.g. Baroni et al., 2007; Cole-Dai et al., 2009; Burke et al., 2019). A new method of measuring ice core sulfur isotopes by multi-collector inductively coupled plasma mass spectrometry (MC-ICPMS) (Burke et al., 2019) has reduced sample size requirements by three orders of magnitude and allows us to generate high time resolution records of $\Delta^{33}S$  throughout the ice core record (McConnell et al., 2017, 2020).

Sulfur isotopes also provide insights into the location of volcanic sources. The $\delta^{34}S$ and $\delta^{33}S$ values in stratospheric sulfate that is deposited in polar ice cores from tropical eruptions are strongly correlated and have a slope of $\ln(\delta^{33}S + 1)/\ln(\delta^{34}S +1)$ = 0.608, rather than the typical terrestrial mass dependent fractionation line with a slope of 0.515 (Burke et al., 2019). This

deviation in slopes results in non-zero $\Delta^{33}S$ values that are strongly correlated with $\delta^{34}S$. In contrast, local, high latitude, stratospheric eruptions can deposit a large amount of tropospheric sulfur with $\Delta^{33}S = 0$, which mutes the overall S-MIF signal preserved in the ice core, and results in $\delta^{34}S$ and $\Delta^{33}S$ that are not strongly correlated. These differences in the relationship between $\delta^{34}S$ and $\Delta^{33}S$ allow us to distinguish between sulfate peaks containing purely stratospheric sulfate (such as tropical eruptions) and those with a tropospheric component (such as extratropical eruptions).

Determining which sulfate peak in ice cores corresponds to the Toba eruption is a challenging task (Svensson et al., 2013). The YTT has been dated using $^{40}Ar/^{39}Ar$ techniques giving ages of $75.0 \pm 0.9$ ka (1 σ) (Mark et al., 2014), $73.88 \pm 0.32$ ka (1σ) (Storey et al., 2012) and $73.7 \pm 0.3$ ka (1σ) (Mark et al., 2017), with the final estimate being a recalculation of the dates from the data from Mark et al., (2014) and Storey et al., (2012). Based on these $^{40}Ar/^{39}Ar$ ages, nine bipolar peaks in sulfate concentration were identified in ice cores within the age uncertainties (Fig. 1; Svensson et al., 2013). These bipolar volcanic ties were correlated between cores in both hemispheres by annual layer counting between the peaks in volcanic sulfate (Svensson et al., 2013).

Although tephra geochemistry is the best way to unambiguously identify an eruption in the ice core, S isotopes can help unravel important information about eruption style and location which can be used in the absence of tephra to understand eruption source. Here we measure sulfur isotope composition of sulfate from two Antarctic ice cores for these nine Toba candidate peaks, as well as two additional sulfate peaks within the Toba $^{40}Ar/^{39}Ar$ age uncertainty window. Using these isotopic measurements we can determine whether the eruption plume reached the stratosphere, and gain insights into their source location (high latitude or tropical). We compare these S isotope results to ice core results from more recent eruptions over the last 2500 years to identify the most likely Toba candidate(s) and provide further constraints on the timing of the eruption and relation to other key paleoclimate records.

## 2 Methods

Ice core samples were collected from two EPICA Antarctic ice cores; Dome C (EDC) (75°06' S; 123°21' E, 3233 m), drilled from 1999–2005 and Dronning Maud Land (EDML) (75°00' S, 00°40' E, 2892 m), drilled from 2000–2006 (Severi et al., 2007). All nine of the Toba candidate peaks of varying magnitude, shown in Fig. 1, have been previously identified in both EDML (Antarctic) and NGRIP (Greenland) cores (Svensson et al., 2013, their Table 2). Five of the candidate peaks have also been identified in the EDC core sulfate record. The T4 peak is not shown in EDC on Fig. 1 as it does not have a peak in sulfate concentration but it is identified by electrical conductivity measurement (Svensson et al., 2013). Samples from EDC included Toba candidates T1, T2 and T3 and an additional peak positioned between T4 and T5 (T4b). The EDML core was sampled for all nine Toba peaks identified by Svensson et al., (2013), as well as a small peak prior to T1 (Pre-T1). Sampling resolution varies across the individual peaks depending on their sulfate concentration. In general, background samples

represent 4–8 years of time whereas samples across the highest concentration regions of the sulfate peaks are 1–2 years. Due

to diffusion these peaks have broadened and thus cover more time than the initial deposition event (Barnes et al., 2003).

Multiple samples were taken across each of these peaks and include pre-event background samples. Where possible, samples

containing a minimum of 20 nmol sulfate were taken, allowing for repeat isotopic measurements (c.f. Burke et al. 2019).

During sampling the outside of ice samples was scraped clean to remove any potential contaminant derived from drilling

fluids.

Sulfate concentration of the samples was determined by ion chromatography using a Metrohm 930 Compact IC Flex using

600 µl aliquots of sample. Once the concentration of the samples was determined, the volume of sample required to give 20

nmol of sulfate was dried down in Savillex PFA Teflon vials on a hotplate. These volumes ranged from 14 ml for

background samples to 2 ml for the highest sulfate concentration samples. Once dry, samples were re-dissolved in 70 µl 0.01

% volume distilled HCl. The sulfate was then extracted from the samples using column chemistry following the procedure

detailed in Burke et al., (2019). For each set of 12 samples, a secondary river water standard, Switzer Falls (Burke et al.,

2018; 2019), and a total procedural blank were also put through columns for analysis. After sulfate is isolated by column

chemistry, the triple sulfur isotope composition ($^{32}$S, $^{33}$S and $^{34}$S) were measured using a Neptune Plus MC-ICPMS (Paris et

al., 2013; Burke et al., 2019) in the STAiG laboratory at the University of St Andrews. Samples and standards were

measured twice during a single analytical session, along with total procedural blanks. The reproducibility of the Switzer

Falls standard over the course of this study was $\delta^{34}$S = 4.15 ± 0.15 ‰ and $\Delta^{33}$S = -0.02 ± 0.17 ‰ (n = 25, 2 s.d.), in

agreement with ratios from previous studies of $\delta^{34}$S = 4.17 ± 0.11 ‰ and $\Delta^{33}$S = 0.01 ± 0.10 ‰ (Burke et al., 2018, 2019).

Procedural blanks had 0.24 ± 0.12 nmol S with a $\delta^{34}$S = 4.32 ± 2.67 ‰ (n = 18, 2 s.d.), and were used to blank correct all

measured ratios.

Background sample(s) from immediately before the peak in volcanic sulfate were used in conjunction with concentration

data to determine the fraction of volcanic sourced sulfate for each sample and the isotopic compositions of the volcanic

fraction ($\delta^{34}$S$_{volc}$ and $\delta^{33}$S$_{volc}$) following Baroni et al., (2007):

$$\delta\ ^{x}S_{volc} = \frac{\delta^{x}S_{meas} - f_{background} \cdot \delta^{x}S_{background}}{f_{volc}} \tag{3}$$

Finally, the $\delta^{33}$S$_{volc}$ and $\delta^{34}$S$_{volc}$ values were used to calculate $\Delta^{33}$S$_{volc}$, with uncertainty propagated throughout using a Monte

Carlo routine. In the case of T1 in EDC, background samples were not available, and as such an average was taken of all

background samples from the EDC core over the Toba time period. Unless stated, we have only considered the $\delta^{34}$S$_{volc}$ and

$\delta^{33}$S$_{volc}$ from samples with volcanic fractions ≥ 65% (Burke et al., 2019; Gautier et al., 2018), as the samples with smaller

volcanic fractions have propagated errors that are prohibitively large to interpret.

## 3 Results

Due to sulfate diffusion in the ice cores, volcanic peaks broaden and reduce in amplitude at depth (Barnes et al., 2003). To account for this diffusion effect we consider the area represented under the sulfate peaks to calculate the total deposition of sulfate for a given event (Sparks et al., in review). The background rate is calculated using the running median over 200 years centred on the peak, and the total deposition is calculated by integrating over the flux from a given peak. The flux is calculated as the product of the concentration in a slice of ice and the snow accumulation rate. However, as the input data is by depths rather than ages, we multiply by the reciprocal of the annual layer thickness at the depth of the slice. As this annual layer thickness is derived from the age model (Veres et al., 2013), the flux is corrected for thinning during the calculation. The deposition of sulfate for the Toba candidates range from 26–133 mg m$^{-2}$ in EDC and 27–424 mg m$^{-2}$ in EDML (Fig. 2, Table S1). In the EDC core over the last 100 ka, 335 sulfate peaks have total depositions >20 mg m$^{-2}$, considered to be the threshold for magnitude 6.5 or greater events (Sparks al., in review). In this ranking, T3 is the 7[th] highest sulfate deposition (133 mg m$^{-2}$) in the last 100 ka. All of the other Toba candidates in EDC are ranked lower with next largest peak, T1 (54 mg m$^{-2}$) ranked 67[th], 1 place above the 1815 CE Tambora eruption. T2 is also in the top 100 peaks with a deposition of 47 mg m$^{-2}$.

Sulfate data are not available from the top of the EDML core however sulfate data is available for the B32 core, drilled 2 km away from EDML (Severi et al., 2007; Göktas, 2008). Total deposition calculations using the FIC (fast ion chromatography) data from 800 CE to 76 ka rank T2 as the highest peak by deposition (424 mg m$^{-2}$), followed by T4 (240 mg m$^{-2}$) as the 5[th] highest. T1 (88 mg m$^{-2}$), T3 (103 mg m$^{-2}$), T6 (96 mg m$^{-2}$) and T9 (86 mg m$^{-2}$) all rank within the top 50 sulfate peaks in the core over this period. These peaks all have a higher deposition than the 1257 CE eruption of Samalas which deposited 74 mg m$^{-2}$ of sulfate to the B32 core. The same peak in different Antarctic ice cores can have vastly different flux estimates. For instance, for T2 the total deposition to the EDC ice core compared to EDML is much lower (46 mg m$^{-2}$ vs 424 mg m$^{-2}$). This disparity is likely due to preservation issues due to the low accumulation rate at EDC site. This issue of preservation has been shown before; for example some cores from the EDC site do not record the 1815 CE Tambora event (Gautier et al., 2016).

Of the 14 peaks measured in EDC and EDML, 13 had samples with non-zero $\Delta^{33}S$ values (Fig. 2). Of all the peaks measured in this study, T1, T2, and T3 had the highest magnitude S-MIF signals (<-0.9 ‰). All of the data measured from these three peaks in both cores are plotted for comparison in Fig. 3, as well as T4 from EDML since it is also considered to be a more likely Toba candidate than T5–T9 (Svensson et al., 2013; Polyak et al., 2017). Results for the other Toba peaks measured in this study are shown in supplementary figures S1–6.

T1 in EDC shows a large decrease in volcanic $\delta^{34}S$ to -48.2 ‰. A negative volcanic $\Delta^{33}S$ of -4.75 ‰ was also measured in the EDC T1 samples, but no positive MIF values were measured. In EDML, T1 shows a negative excursion in $\delta^{34}S_{volc}$ to -29.4 ‰. Unlike the EDC samples for T1, both a positive and negative volcanic MIF excursion was measured in EDML with the $\Delta^{33}S$ signal evolving from background to +1.34 ‰ then dropping to -3.08 ‰. These are the largest magnitude volcanic MIF signals measured in this study for the Toba candidates in each of the cores.

T2 in EDC initially shows little change in measured $\delta^{34}S$ before decreasing to -13.6 ‰, with a $\delta^{34}S_{volc}$ of -21.9 ‰. The background corrected data show an initial positive $\Delta^{33}S$ of +0.57 ‰, and after this the remainder of the samples record a negative MIF signal, with $\Delta^{33}S_{volc}$ falling to -3.41 ‰. The volcanic $\delta^{34}S$ and $\Delta^{33}S$ signals in EDML show a very similar evolution to their EDC counterparts with a decrease in $\delta^{34}S$ to -6.46 ‰ before returning to background levels. Both positive and negative volcanic MIF signals are recorded, increasing up to +0.74 ‰ before reducing to -1.72 ‰.

The measured $\delta^{34}S$ from T3 in EDC decreases from a background value of +16.3 ‰ to -0.8 ‰ before returning to +16.1 ‰. The volcanic MIF evolution follows a trend from positive (+0.62 ‰) to negative (-0.93 ‰). The measured $\delta^{34}S$ signal for T3 in EDML decreases from background level of +16.2 ‰ to +9.9 ‰ during the eruption before increasing again. In contrast to T3 in EDC, the EDML samples have preserved only a positive volcanic MIF signal of +0.99 ‰.

Of the remaining peaks, the peak prior to T1 and T4–T9 in EDML all have one or more samples with a non-zero $\Delta^{33}S$, though smaller in magnitude than measured in T1, T2 and T3 (see Table S1 and figures S1–6). The additional peak between T4 and T5 sampled from EDC had the lowest sulfate deposition of the Toba candidates in the sampled peaks from EDC at 26 mg m$^{-2}$. This is the only peak in this study where all samples had $\Delta^{33}S$ values within 2σ of 0 ‰.

## 4 Discussion

### 4.1 Preservation of S-MIF in Toba candidate events from two Antarctic ice cores

The presence of non-zero $\Delta^{33}S$ (outside of ± 0.17 ‰ uncertainty, see *Methods*) in volcanic sulfate from ice cores suggests that the sulfate deposited from these eruptions came from the stratosphere at altitudes at or above the ozone layer where it could interact with UV radiation (Savarino et al., 2003). Of the 14 Toba candidates measured in this study across both EDC and EDML cores, 13 showed a non-zero stratospheric MIF signal (Fig. 2). The only event that had a zero S-MIF was the small peak between T4 and T5 sampled in EDC. This result suggests that the sulfur associated with this volcanic eruption did not make it as high as the ozone layer and it was likely a local eruption, which could also account for the lack of a corresponding sulfate peak in EDML (Fig. 1).

For the three peaks that were measured in both EDC and EDML (T1–3), there is good agreement in the S isotope records between the two cores (see Fig. 3). The main difference is that the EDC $\delta^{34}$S and $\Delta^{33}$S are consistently more negative than

those measured in EDML. This could be due to an increased influence of marine sulfate ($\delta^{34}$S ~20‰, Paris et al., 2013) at EDML that is not adequately accounted for in the background correction, which could be influenced by factors such as diffusion. There are also slight differences between the cores in the preservation of the positive or negative S-MIF across individual events. For example, only the negative S-MIF signature is preserved in T1 in EDC, and only the positive S-MIF signature is preserved in T3 in EDML, whereas T1 in EDML and T3 in EDC show both positive and negative S-MIF.

Previous studies that have measured S isotopes in multiple samples across a volcanic sulfate peak in ice or snow have shown that the $\Delta^{33}$S evolves from positive to negative values (e.g. Baroni et al., 2007; Cole-Dai et al., 2009; Burke et al., 2019). Thus, the initial sulfate that is deposited on ice sheets following a volcanic eruption inherits a positive S-MIF signal in the stratosphere, and this sulfate is isolated (in time or space) from the residual sulfate that is deposited later with a negative S-MIF signature (Baroni et al., 2007; Gautier et al., 2018). Mass balance should require that the sum of the positive and

negative anomalies should balance out. However at low accumulation rate sites, such as Dome C, occasionally only the positive or negative component is preserved in the core (Gautier et al., 2018, 2019). This loss of a proportion of the sulfate is perhaps not surprising in these cores given that at low accumulations sites like Dome C, entire sulfate peaks (such as the 1815 Tambora eruption) might be missing from an individual core (Gautier et al., 2016). Given that some volcanic sulfate may not be preserved in these Toba candidate events, and that sulfate diffuses in the ice cores, the isotope compositions

measured in the Toba candidate peaks should be viewed as minimum magnitude $\Delta^{33}$S values. Due to the low accumulation rate at Dome C the calculated sulfate depositions are minimum estimates.

As most of the Toba candidates measured in the cores have a stratospheric S-MIF signal, the Toba eruption cannot be immediately identified purely on the presence or absence of S-MIF. However, recent work by Lin et al., (2018) shows a

correlation between $\Delta^{33}$S and cosmogenic $^{35}$S in modern aerosols which suggests that sulfate formed at higher altitudes has a greater positive $\Delta^{33}$S. Although the exact photochemical process that imparts the S-MIF is not fully understood (see e.g. Gautier et al., (2018) for a discussion), the implied relationship between altitude and $\Delta^{33}$S suggests that the magnitude of the $\Delta^{33}$S recorded in ice cores may scale with a volcanic eruption's plume height. Due to isotopic mass balance, the large negative $\Delta^{33}$S signals measured for T1 and T2 in EDC would also have a complementary large positive $\Delta^{33}$S signal which

was not deposited and preserved due to the low accumulation at the site. In turn, these large positive signals would be indicative of exceptionally large volcanic plume heights.

We can further narrow down potential Toba candidates by considering the $\delta^{34}$S signal measured across these events as well. Given the low latitude location of the Toba caldera, any sulfate deposited in Antarctica from the Toba eruption would have

to be delivered via the stratosphere from heights at or above the ozone layer. Due to the difference in the $\ln(\delta^{33}S +1)$ vs $\ln(\delta^{34}S +1)$ slope from stratospheric and terrestrial sulfate (0.608 vs 0.515 respectively), non-zero $\Delta^{33}$S in stratospheric

sulfate is strongly correlated with $\delta^{34}$S (Burke et al., 2019). In contrast, if some tropospheric sulfate is present in the ice core, for example from an extratropical eruption proximal to Antarctica (even if it was a stratospheric eruption), the sulfur isotope compositions measured in ice would not exhibit a strong positive correlation between $\Delta^{33}$S and $\delta^{34}$S. This is because $\Delta^{33}$S in

the tropospheric sulfate would be zero, and its $\delta^{34}$S value would reflect the isotopic composition of the erupted sulfur gases (Burke et al., 2019). Indeed, for the extratropical eruption of Katmai/Novarupta (Alaska) in 1912, the sulfur isotope compositions measured in Greenland showed a weak negative correlation between $\Delta^{33}$S and $\delta^{34}$S (Burke et al., 2019).

Of the stratospheric candidates, T1, T2 and T3 all show a strong positive correlation between $\delta^{34}$S and $\Delta^{33}$S in both cores

suggesting that the sulfate comprising these peaks was purely stratospheric in origin. This implies they are most likely to be from a tropical eruptions source and therefore are the best candidates for the Toba eruption. It should be noted however that these sulfate peaks could also have been deposited by an extratropical event with a low tropospheric sulfur load. Overall, the linear regression (York et al., 2004) for these three events in two cores gives a slope of $0.106 \pm 0.003$ ($1\sigma$) with an $r^2$ value of $0.964$, and similar to the slopes reported by Gautier et al (2018) of $0.09 \pm 0.02$ and Burke et al (2019) of $0.089 \pm 0.003$ for

tropical eruptions measured in ice cores (Fig. 4a).

The remaining Toba candidates either did not have multiple samples with enough volcanic sulfate ($f_{volc} \geq 0.65$) to investigate correlations (Pre-T1, T5, T7, T8, T9) or they did not show strong positive correlations between $\delta^{34}$S and $\Delta^{33}$S (T4 and T6). Both T4 and T6 in EDML have only weak statistical correlations between $\delta^{34}$S and $\Delta^{33}$S with slopes of $0.081 \pm 0.063$ and

$0.037 \pm 0.018$ respectively with weighted $r^2$ values of $0.088$ (T4) and $0.065$ (T6). This indicates that a substantial proportion of the sulfate deposited on the ice sheet originated from below the ozone layer. Given the presence of MIF in both of these sulfate layers, we can conclude that the eruptions were large enough to reach the stratosphere at or above the ozone layer, but likely they were extratropical. In the case of T4, this could in part help to explain why this sulfate peak is not obviously apparent in EDC if a large portion of the sulfate in EDML was deposited directly via the upper troposphere/lower

stratosphere. Although it is possible for large extratropical eruptions to deposit sulfate at both poles (Baines et al., 2008; Toohey et al., 2016) the bipolar attribution of T4 and T6 (and associated chronological constraints on ice core age models) should be investigated further for these events by analysing high resolution sulfur isotope compositions from Greenland ice cores.

The combination of an incomplete geological record of past volcanism along with large uncertainties in dating of geological samples mean that it is not possible to unambiguously attribute a volcanic event with a sulfate deposition event in ice cores in the geological past unless there is a tephra confirmation of the source. We take the approach that given the age estimates of Toba, we can investigate all possible candidates within the age uncertainty and rule out candidate eruptions if they have a muted or weak MIF signal. Although we cannot rule out the possibility that other eruptions deposited these sulfate peaks,

providing that the dates of the YTT are accurate, and that it emitted substantial sulfur, then at least one of the candidates we

investigated is very likely to be Toba. Using the VOGRIPA database (Crosweller et al., 2012; Brown et al., 2014) we have identified other volcanic events over the age range of T1–9 (when considered on the AICC2012 age model). These volcanic events and their associated dates are detailed in the supplementary Table S4. There are 9 events with VEI ≥ 6 at around 74 ka in VOGRIPA, however they often have large age uncertainties associated with the eruption dates (over 10 ka). Thus, there are many more peaks in addition to T1–T9 in the ice core record that could have been deposited by these eruptions. One of the few with a smaller error is a VEI 6 eruption from the Coatepeque Caldera dated to $72 \pm 2$ ka (Rose et al., 1999). However, the Toba eruption is the largest of the candidate eruptions over the age range encompassed by T1–T9, and in order to find an eruption with significantly larger S deposition than those considered here at EDC, one would have to extend the search to 79.5 ka, which is well outside the uncertainty in the age of the Toba eruption. Therefore, unless the YTT age and its uncertainty is not accurate or it had negligible sulfur emission, both highly unlikely, then the YTT must have resulted in at least one of the T1–T9 candidates.

## 4.2 Toba sulfur loading and comparison to Common Era eruptions

Sulfur mass independent fractionation in ice core sulfate has been measured in numerous volcanic events over the last 2600 years (Fig. 5, Savarino et al., 2003; Baroni et al., 2007; Baroni et al., 2008; Cole-Dai et al., 2009; Gautier et al., 2018; Burke et al., 2019; Gautier et al., 2019). The negative S-MIF signals measured in T1 and T2 are the largest magnitude volcanic S-MIF signals in our study, and in T1 in EDC has the most negative $\Delta^{33}S$ value (-4.75‰) ever recorded in ice core sulfate (Fig. 5).

For further comparison to Common Era eruptions, we compared the total sulfate deposition for T1, T2 and T3 (see Fig. 2a) to the sulfate deposition of major eruptions in the Common Era. This is achieved by first comparing the deposition from Common Era eruptions in the EDC and EDML ice core records to the calculated total Antarctic deposition for the same Common Era events from the compilation of Antarctic cores in Sigl et al., (2015). For each of the events identified in both cores (Table S5) we averaged the sulfate deposition and compared to the total Antarctic deposition. The correlation ($r^2$ = 0.77; Fig. S7) between these deposition masses suggests we can use this relationship to scale the average sulfate deposition of the Toba candidates in EDC and EDML to approximate a total Antarctic deposition for these events. We have used sulfate concentration data for NGRIP from Svensson et al., (2013) and accumulation data from Kindler et al., (2014) to determine the sulfate deposition for T1, T2 and T3 in the NGRIP core. This is subsequently scaled to a total deposition over Greenland using the relationship established by Toohey and Sigl, (2017). These total depositions over Greenland and Antarctica are used to calculate the stratospheric sulfur loading following the method detailed in Toohey and Sigl, (2017). Scaling the sulfate flux in this manner, we calculate stratospheric sulfur loadings of 154, 233 and 72 Tg S for T1, T2 and T3 respectively, all of which are greater than the stratospheric loadings due to the 1257 Samalas eruption of 59.4 Tg S (Toohey and Sigl, 2017).

These sulfur emissions fall within the broad range of estimates of sulfur loading for the Toba eruption (e.g. 33–1750 Tg from petrological and volume estimates (Chesner and Luhr, 2010; Costa et al., 2014) and previous ice core estimates of T1 (Zielinski et al., 1996)), but are on the lower range of estimates. These results suggest that either there is an incorrect assumption in the scaling of revised erupted volume calculations leading to the large estimates of 850–1750 Tg S (Costa et al., 2014) or a large portion of the erupted sulfur either did not make it into the stratosphere or it was rapidly removed. Both scenarios suggest that despite the massive erupted volume associated with the YTT eruption, the stratospheric sulfate loading was not orders of magnitude larger than more recent eruptions within the Common Era, but may have still been 2–4 times greater than that of the 1257 Samalas eruption.

Estimating the sulfur yield from explosive silicic eruptions is a complex matter and eruption magnitude is only one of several factors that control yield. In silicic magmatic systems, sulfur is partitioned between solid, melt, and fluid phases (Masotta et al., 2016). A major uncertainty in petrological estimates is how much sulfur is stored in exsolved fluids in the magma chamber. This latter form of sulfur may be significant or even dominant when there is strong partitioning of sulfur between melt and fluid phase. Fluid-melt partitioning is mainly influenced by sulfur speciation and hence redox conditions (Scaillet et al., 1998; Binder et al., 2018). Redox estimates for YTT suggest relatively reduced conditions around the Ni-NiO (NNO) buffer (Chesner, 1998). Experimental work of Scaillet et al. (1998) suggested that melts at NNO are unlikely to exsolve significant quantities of sulfur in an eruptible fluid phase, however, more recent experiments of Binder et al. (2018) challenge this and indicate that sulfur is strongly partitioned into the fluid phase for a melt at NNO. To date no experimental study has investigated natural samples of Toba pumice at appropriate temperature, pressure and redox conditions, and so it remains uncertain as to whether a substantial S-rich fluid phase existed prior to eruption and contributed to the sulfur loading. One further issue is that large magma chambers that lead to major explosive eruptions take long periods of time to assemble, as exemplified by Toba with zircon ages spanning 100's of ka (Reid and Vazquez, 2017). Even if a large volume of sulfur rich magmatic fluids were exsolved from the magma, one might expect that over the long period of magma reservoir assembly that significant sulfur was lost to the surrounding hydrothermal system and degassed prior to eruption. The key point is that current petrological estimates of sulfur yield from Toba generally suggest low values (comparable to large common era eruptions such as Tambora and Samalas) consistent with our ice core sulfur flux estimates, although more detailed experimental investigations are essential to confirming this.

One of the key questions that our study of Toba's ice core record raises is: should exceptionally large volume eruptions have a commensurately large ice core sulfur peak, in other words should we expect any relationship between sulfur loading and erupted volume? To investigate this, we considered the sulfur loading relative to eruptive volume (given as a dense rock equivalent, DRE) for a variety of Common Era eruptions (Table 1). These ratios show significant variation between different events, for example the value for 1257 Samalas eruption is 1.5–1.8 whereas the 946 Millennium Eruption (ME) associated with Paektu volcano shows a lower ratio of 0.24. The ratios for the Toba candidates vary from 0.02 up to 0.12 when T1, T2

and T3 are combined, with the upper values the same order of magnitude as the ME. This simple analysis shows that we should not expect large eruptions to necessarily have the largest ice core sulfur peaks. Magma chamber assembly, storage

conditions, and redox will all influence the sulfur budget prior to eruption (Scaillet et al., 1998; Reid and Vazquez, 2017; Binder et al., 2018), while plume dynamics and ice core preservation will impact the sulfur depositional signal (Gautier et al., 2016). On present evidence we would expect Toba to have a S yield a few times larger than Tambora and possibly more if a significant exsolved S fluid phase existed prior to eruption. Our estimates for T1, T2 and T3 sulfur yield are 5.5, 8.5 and 2.5 times greater than the Tambora eruption respectively. Therefore, although Toba is a huge eruption in terms of erupted

material, it is not exceptional in terms of sulfur loading. The low Toba sulfur loading observed in ice core records is consistent with the lower petrological estimates (Chesner and Luhr, 2010) and not unexpected since Common Era events like the Millennium Eruption (Table 1) clearly demonstrate that large erupted volumes do not always lead to large ice core sulfate peaks.

Table 1. Comparison of the sulfur loading estimates derived from ice cores and DRE for volcanic eruptions through time and
the Toba candidates in this study. The sulfur loading estimate for the ME eruption was calculated following the methodology of Toohey and Sigl, (2017) from sulfur deposited in the ice core records (Sigl et al., 2015; Sun et al., 2014).

| Eruption | Sulfur loading (Tg S) | DRE (km$^3$) | Ratio (Tg S km$^{-3}$) | References |
|---|---|---|---|---|
| Tambora 1815 CE | 28.1 | 41 | 0.69 | (Kandlbauer and Sparks, 2014; Toohey and Sigl, 2017) |
| Samalas 1257 CE | 59.4 | 33–40 | 1.5–1.8 | (Toohey and Sigl, 2017; Vidal et al., 2015) |
| Millennium Eruption ~946 CE | 5.7 | 24 | 0.24 | (Horn and Schmincke, 2000) |
| T1 | 154 | 3800 | 0.04 | (Costa et al., 2014) |
| T2 | 233 | | 0.06 | |
| T3 | 72 | | 0.02 | |
| T1+T2 | 387 | | 0.10 | |
| T1+T2+T3 | 459 | | 0.12 | |

**4.3 Toba eruption dynamics**

The atmospheric sulfur loading from an eruption depends on the eruption dynamics. Caldera forming eruptions
can often erupt in two phases. The first phase is a Plinian eruption from a single vent producing a high, buoyant

plume. Then when the cone collapses, multiple vents can form leading to pyroclastic flows and co-ignimbrite plumes (Wilson, 2008). In the case of Toba the marine ash layer in the Indian Ocean displays a coarse lower layer and a fine grained upper layer (Ninkovich et al., 1978). The lower layer was interpreted by Ninkovich et al (1978) as a Plinian deposit, but no basal Plinian deposit has been yet identified beneath the ignimbrite surrounding the

Toba caldera leading the interpretation that the distal ash deposits are co-ignimbrite in origin (Chesner, 2012). Co-ignimbrite plumes can transport sulfur compounds into the stratosphere and can reach plume altitudes similar to Plinian plumes (Engwell et al., 2016) and deposit ash over a wide area (Chesner, 2012). In addition, over-plumes forming above the main plume for events on the scale of Toba could potentially extend the total plume height to ~60–70 km (Costa et al., 2018).


The Toba ash fall deposit is consistent with an exceptional plume height in two ways (Ninkovich et al., 1978; Liang et al., 2001). First is the large area of ash dispersal covering much of the northern Indian Ocean and Indian sub-continent and as far as East Africa in the west and extending to the South China Sea to the northeast (Lane et al., 2013). Second is the exceptional coarse grain size of the Toba deposit which has a median grain size about 8

times greater than the tephra deposits of Tambora 1815 and the Late Bronze Age Santorini tephra deposits at about 500 km from source (Ninkovich et al., 1978; Kandlbauer et al., 2013) and 2.5 times greater than the Campanian tephra deposits at about 1700 km from source (Engwell et al., 2014). These features can be explained by a giant eruption cloud of exceptional size. The height of the Toba plume can be constrained by modelling (c.f. Baines and Sparks, 2005; Costa et al., 2014). These results indicate cloud heights exceeding 40 km, although

these estimates should be caveated by the large uncertainties in the input parameters, such as eruption duration (Oppenheimer, 2002).

When we compare the S-MIF signals for our Toba candidates, particularly T1, T2, and T3, to previous studies of Common Era events we find that the larger magnitude events, such as Samalas and Tambora, have larger

magnitude MIF signals (Fig. 5). Independent geological estimates of eruption plume height are available for numerous Common Era eruptions (Aubry et al., 2021), determined by a range of methods including using cloud positions from satellite sensors for the $SO_2$ injection altitude (Guo et al., 2004) and modelling lithic clast dispersal for the plume top height (Sigurdsson and Carey, 1989). There is a positive correlation between the maximum $\Delta^{33}S$ value measured and plume top height for Common Era eruptions that have high resolution ($\geq 5$

samples/peak) S isotope measurements from ice cores (Burke et al., 2019) or snow pits (Baroni et al., 2007) (Fig. 6). If this relationship is extrapolated to the range of MIF signals measured for T1 and T2, we would anticipate

the plume top height to be in excess of 45 km, similar to the model estimate of the Toba eruption of ~ 42 km from Costa et al., (2014). A more appropriate eruptive parameter to compare with the maximum $\Delta^{33}S$ is the $SO_2$ dispersion height, as this is the altitude of the sulfur plume and thus is likely be the altitude at which the S-MIF is

inherited. However the $SO_2$ dispersion height is not as well constrained for past eruptions as the plume top height (Aubry et al., 2021). For the Agung and Pinatubo eruptions, we have used the $SO_2$ dispersion altitudes from the literature, compiled in the IVESPA database (Aubry et al., 2021; ivespa.co.uk). For the Tambora and Samalas eruptions we have calculated the $SO_2$ dispersion height using the ratio of $SO_2$ height to plume top height for the Pinatubo eruption (0.64). With this data we place the $SO_2$ dispersion height for Toba at over 30 km (Fig.6).


Although this tentative relationship between $\Delta^{33}S$ and plume height supports the conclusion of an altitude dependence of S-MIF by Lin et al., (2018), there are some important caveats to note. For instance, the maximum magnitude of the $\Delta^{33}S$ measured in the ice will depend on the sampling resolution and preservation in the core. We have tested the impact of decreasing sampling resolution by averaging samples from Tambora and Samalas

from Burke et al. (2019).  In these two instances, the lower sample resolution reduces the $\Delta^{33}S_{volc}$ signal slightly as expected, but the average remains within error of the measured maximum S-MIF value (see Fig. S8). This exercise further illustrates the importance of maximizing sampling resolution, which is made possible by measurement with MC-ICP-MS (Burke et al., 2019).

Furthermore, our estimates of plume height or $SO_2$ dispersion height are based on a tentative relationship from only a handful of Common Era eruptions with large uncertainties associated with whether the S-MIF measurements captured the maximum magnitude of the S-MIF signal (see Supplementary Info, Fig. S8). Thus this relationship should be further investigated and validated. One such method that could provide additional information on plume height is sulfate oxygen MIF measurements ($\Delta^{17}O$), since O-MIF in OH in the stratosphere varies with altitude (Zahn et al., 2006) which can then be

inherited by sulfate during oxidation (Gautier et al., 2019).

Using annual layer counting, T1 and T2 are separated by ~80–90 years in the ice core record (Svensson et al., 2013). Given this close spacing and the large negative S-MIF signals for both peaks, there is an alternative interpretation that T1 and T2 and possibly even T3 are all pulses of the YTT. Such an interpretation is supported by recent research on comparable

supereruptions which indicate that these eruptions can be more complex multi-episode events. The Huckleberry Tuff erupted from Yellowstone caldera is a comparable magnitude eruption to Toba and detailed field studies (Swallow et al., 2019) show that there were three episodes with gaps of weeks to months occurring between the first two episodes, and years to decades between the second two. The explanation of multiple pulses reflects the disturbance of multiple spatially separated magma

chambers. Petrological and geochemical evidence for supereruptions having more complex prolonged eruptive histories,
disturbing multiple chambers and involving magma assembly over decades to centuries is emerging in other examples (Druitt et al., 2012; Shamloo and Till, 2019; Pearce et al., 2020). Thus, the assumption that such Toba and Yellowstone magnitude eruptions are single geologically instantaneous events may be wrong and opens up the option of assigning multiple closely spaced sulfate peaks to Toba, since the bulk rock estimates may suggest a single large event because of the temporal resolution of geological ages, while the ice cores could record multiple sulfur peaks. In addition, this would
increase the total sulfur loading estimate for Toba to over 6.5 times the size of Samalas (387 Tg S; T1 + T2) or as much as 7.7 times that of Samalas (459 Tg S; the sum of T1, T2 and T3).

## 4.4 Impact and timing of the Toba eruption

Given the scale of the Toba event, we consider the potential impact of the eruption on global climate. Over the last ice age,
millennial duration cycles of abrupt changes in climate in the Northern Hemisphere are recorded in paleoclimate records (Broecker, 2000). The estimated dates for the Toba eruption coincide with the transition into one of these cold periods, Greenland Stadial 20 (GS-20); however it is unclear as to whether the Toba eruption caused this Northern Hemisphere cooling or if it was already underway (Baldini et al., 2015).

Our data allow us to narrow down the Toba candidates and hence the timing of the eruption and its relationship to GS-20. As T1 and T2 show the largest magnitude S-MIF signals we consider them to be the best candidates for the Toba eruption, which places them on the transition into GS-20 (see Fig. 7 below; Svensson et al., 2013). This relative timing implies that the Toba eruption was not a trigger for the Northern Hemisphere cooling.

We can also use the ice core records to place further constraints on the absolute timing of the Toba eruption. There are two potential age models for NGRIP records over this time interval: GICC05modelext (Wolff et al., 2010b) and AICC2012 (Veres et al., 2013). Although the GICC05modelext and AICC2012 timescales are the same over the last 60.2 ka, the transitions between stadial and interstadial periods can differ on the two age scales by up to a couple of millennia by 100 ka (Veres et al., 2013). Since the AICC2012 age model takes into consideration the volcanic bipolar synchronization over this
particular time period from Svensson et al., (2013), studies that compare records from both hemispheres are encouraged to use the AICC2012 age model to not introduce any artificial leads and lags in records, whereas the GICC05modelext continues to be used when considering only Greenland records. These differences in age models put T1 and T2 in NGRIP at 74.06 and 74.16 ka on the GICC05modelext, and 73.66 and 73.75 ka on the AICC2012 age model respectively.

While these differences in ages between the two age models are well within the uncertainties of the ice core age models around this time (~1500 years, Veres et al., 2013), we can further improve on the estimates of the absolute age of the T1 and

T2 events through comparison of the Greenland $\delta^{18}O$ records with speleothem records of the Asian monsoon (e.g. Wang et al., 2001; Chen et al., 2016; Du et al., 2019; Corrick et al., 2020). Speleothems can be radiometrically dated using U-Th techniques with uncertainties of <200 years around this time (Du et al., 2019), and a systematic analysis of the timing of

transitions out of stadial events in Greenland and weak monsoon events recorded in speleothems over the past 45 ka has shown these transitions are synchronous within 189 years (Adolphi et al., 2018). Corrick et al., (2020) provide a comprehensive global compilation of 63 published speleothem records, providing dates of the interstadial transitions of $71,594 \pm 230$ years for GI-19.2 and $75,583 \pm 248$ years for GI-20c. In comparison, using the AICC2012 age model we place these transitions at 72,142 years and 75,876 years with the GICC05modelext giving values of 72,340 years and 76,440 years

(Rasmussen et al., 2014). Further interrogation of these global records to determine the onset of GS-20 may further improve our estimates for the ages of T1 and T2.

However, it is possible to improve age constraints using detailed speleothem records over the GS-20 transition. Previous work used the correlation between Greenland $\delta^{18}O$ records and speleothem $\delta^{18}O$ records with change point analysis to

investigate the timing of Toba candidates (Du et al., 2019). Using a speleothem record of $\delta^{18}O$ from Yangkou which has the highest resolution $\delta^{18}O$ and best age control over this time period, Du et al. suggested a shift in the GICC05modelext by 150 years. However, because the duration of the Greenland stadial is 260 years longer than the duration of weak monsoon event (1760 years v 1500 years respectively, determined using the NGRIP depths and ages from Rasmussen et al., 2014), choosing whether to apply the change point analysis to the initiation of GS-20 or the initiation of the Greenland Interstadial 19 impacts

the estimated ages of the Toba candidate events (Du et al., 2019). The duration of GS-20 on the AICC2012 age model (1550 years) on the other hand is more similar to the duration of the weak monsoon event recorded in the Yangkou speleothem (1500 years), and both the transitions into and out of GS-20 on the AICC2012 age model are within the ~100–200 year U-Th age uncertainty of the weak monsoon event transitions reconstructed in the speleothem (Fig. 7). Thus, without having to apply any shifts to the AICC2012 age model, the absolute ages of the NGRIP $\delta^{18}O$ record on the AICC2012 age model are

consistent with the radiometrically dated speleothem over this time interval. We can then use T1 and T2 on the AICC2012 age model to suggest revised ice core dates of our best candidates for the Toba eruption of 73.65 and 73.77 ka BP, with an uncertainty of about 0.2 kyr based on the uncertainties from the U-Th ages (Du et al., 2019) and any potential leads and lags in the correlation between the Greenland and the speleothems (Adolphi et al., 2018). This range of dates is in agreement with the ages for the YTT from Storey et al., (2012) ($73.88 \pm 0.64$ ka, $2\sigma$) and Mark et al., (2017) ($73.7 \pm 0.6$ ka, $2\sigma$), though with

smaller uncertainties.

**5 Conclusions**

Our data shows that distinct, large magnitude MIF signals can be preserved in volcanic ice core sulfate despite undergoing diffusion for over 70 ka. Given the large magnitude MIF signals recorded in both EDC and EDML and the positive

correlation between $\Delta^{33}S_{volc}$ vs $\delta^{34}S_{volc}$, T1 and T2 are the most likely candidates for the Toba eruption. This in turn places

the Toba eruption between 73.65–73.77 ka, during the transition to Greenland Stadial 20. This would suggest Toba to be an unlikely trigger of the stadial period, although we cannot rule out a role for the eruption in amplifying the transition. Our estimates of sulfate flux deposited in both poles for these events suggest that Toba's stratospheric sulfur loading was 2–4 times greater than to that of the 1257 CE Samalas eruption, which was the largest volcanic sulfur loading over the last 2000 years. Using the sulfate flux data for the EDC and EDML ice cores we find that T2 has a sulfur loading of 233 Tg S, which

is approximately 4 times greater than that of Samalas reported by Toohey and Sigl, (2017) of 59.4 Tg S. The sulfate deposition to the ice cores due to Toba may be small relative to its estimated magnitude. However, the radiometric age constraints of the YTT and bipolar synchronization of the ice cores (Svensson et al., 2013) leave T1, T2 and T3 as the best candidates for the Toba eruption in the ice cores.

Sulfur isotopic composition measurements on ice core sulfate from Greenland identified as the Toba candidates, particularly T1 and T2, would be an essential next step in further constraining the Toba eruption in the bipolar ice core record. In addition to sulfate, tephra can also be used to identify volcanic eruptions in ice cores (Abbott and Davies, 2012). Toba tephra has yet to be identified in either Greenland or Antarctic ice cores (Svensson et al., 2013) despite tephra being identified in Antarctica for other similar or smaller magnitude tropical volcanic eruptions such as Samalas (Lavigne et al., 2013).

However with method improvements in tephra extraction (Iverson et al., 2017; Narcisi et al., 2019) it may be possible to identify the volcanic peak(s) corresponding to Toba and further constrain the timing of the largest volcanic event of the Quaternary.

**Acknowledgements**

The authors are gratefully for the assistance of Helen Innes in sulfur isotope composition measurements and Mirko Severi

and Thomas Aubry for discussions regarding FIC sulfate data and $SO_2$ plume altitudes respectively. We thank Anders Svensson and Jihong Cole-Dai for reviewing this article and providing thorough and comprehensive feedback. This paper also benefitted from discussion at events of the Past Global Changes (PAGES) working group 'Volcanic Impacts on Climate and Society' (VICS).

LC is funded by the University of St Andrews St Leonards 7[th] Century Scholarship (-117THCENT01) in partnership with the IAPETUS Natural Environment Research Council Doctoral Training Partnership. AB is funded by a European Research Council (ERC) Marie Curie Career Integration Grant (CIG14-631752) and NERC Strategic Environmental Science Capital Call (CC082). WH is funded by a UKRI Future Leaders Fellowship (MR/S033505/1). MK was funded through DFG-Project Wo 362 /32-1 and Wo 362/46-1,2 to G. Wörner at GZG, University Göttingen. RSJS is funded by a

Leverhulme Grant (RPG-2015-246) and Leverhulme Emeritus Fellowship (EM-2018-050) and EWW is funded by a Royal Society Professorship.

This work is a contribution to the European Project for Ice Coring in Antarctica (EPICA), a joint European Science Foundation/European Commission (EC) scientific programme, funded by the EU and by national contributions from
Belgium, Denmark, France, Germany, Italy, The Netherlands, Norway, Sweden, Switzerland and the UK. The main logistic support was provided by IPEV and PNRA (at Dome C) and AWI (at Dronning Maud Land). This is EPICA publication number 318.

**Author contribution**

LC, EAD, and SM sampled the EDC ice cores and MK and SK sampled the EDML ice cores. AB and KAM measured S
isotopes from EDML and LC measured S isotopes from EDC with supervision from AB, RCJS, and JWBR. LC and AB analysed isotope data and LC analysed sulfate concentration data, with help from EWW. LC and AB developed the initial interpretation and LC wrote the initial draft manuscript, with early contributions from AB, WH, JS, RSJS, and EWW. AB, RSJS, and MK funded the project through grants (see acknowledgements). AB supervised the project. All co-authors contributed to the writing and editing of the final submitted manuscript.

**Data availability**

The sulfur isotope composition data for EDC an EDML ice core samples is available on the PANGAEA data repository: Crick, Laura; Burke, Andrea; Hutchison, William; Kohno, Mika; Moore, Kathryn A; Savarino, Joël; Doyle, Emily A;
Mahony, Sue H; Kipfstuhl, Sepp; Rae, James W B; Steele, Robert C J; Sparks, R Stephen J; Wolff, Eric W (2021): High-resolution sulfur isotopic composition measurements of volcanic sulfate from Toba candidate eruptions preserved in EDML and EDC Antarctic ice cores. PANGAEA, https://doi.org/10.1594/PANGAEA.933271

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

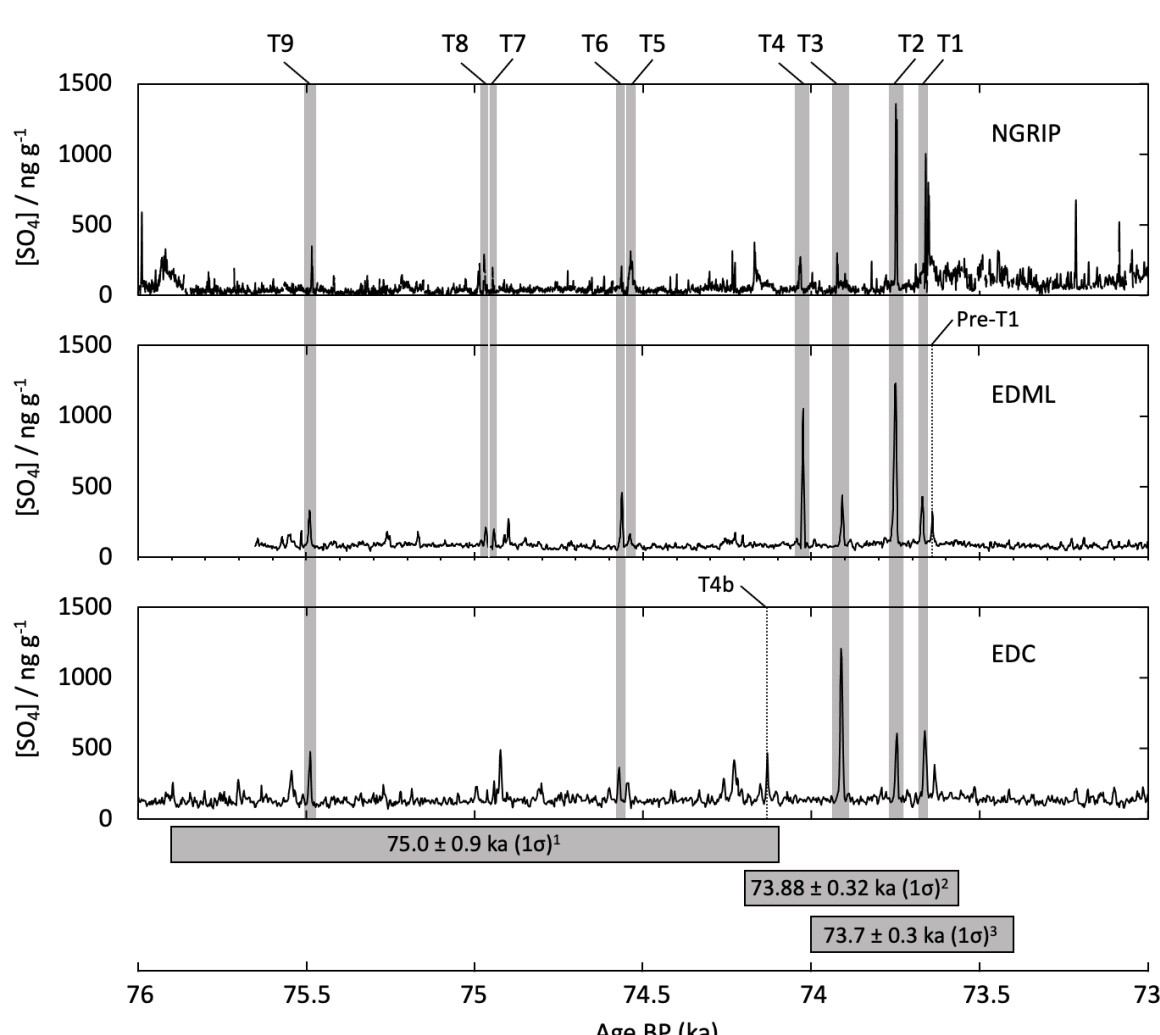

**Figure 1. Toba candidates identified in both Greenland (NGRIP) and Antarctic (EDML, EDC) ice cores by Svensson et al., (2013). Sulfate concentration data from Svensson et al., (2013) [NGRIP and EDML] and Sparks et al., (in review) [EDC]. The AICC2012 ice core chronology (Bazin et al., 2013; Veres et al., 2013) was applied to both Greenland and Antarctic ice cores. Dotted lines indicate**

**the additional peaks measured in EDC and EDML. The age estimates for the Toba eruption from 1) Mark et al., (2014), 2) Storey**

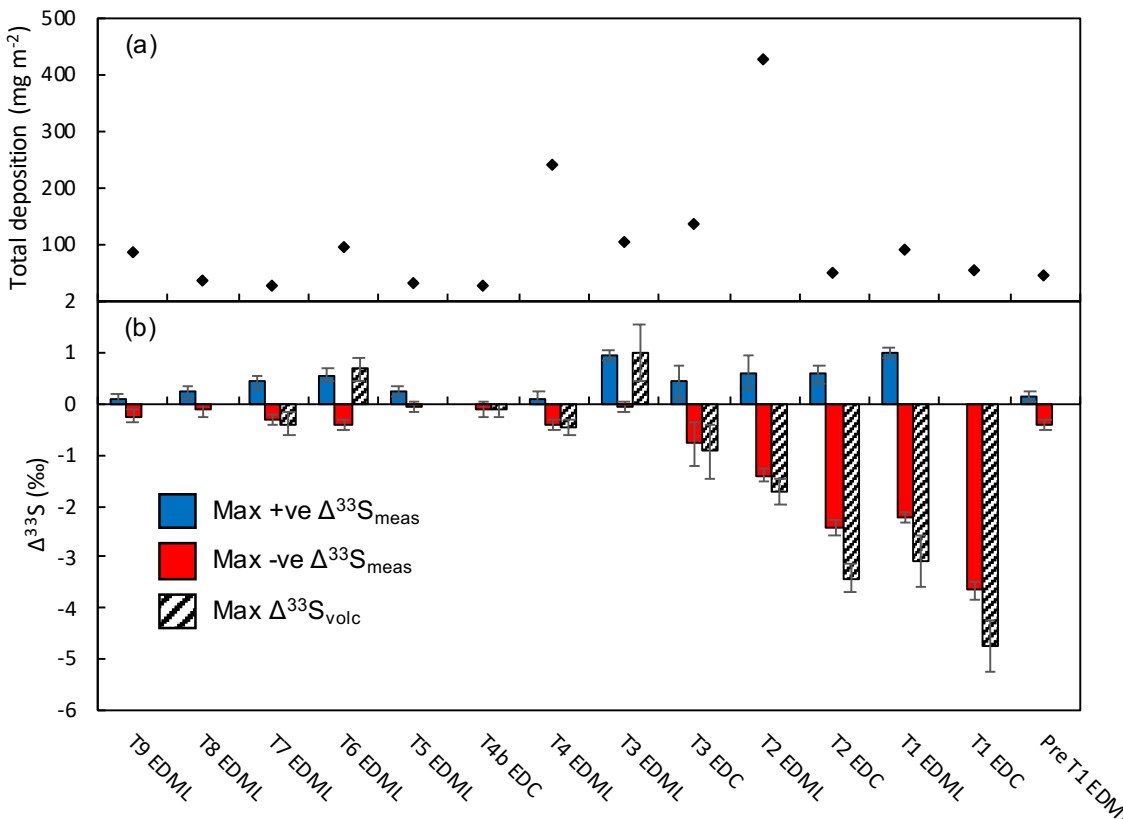

et al., (2012) and 3) Mark et al., (2017) are represented by horizontal bars indicating 1σ for each date. Note that the stated uncertainty in AICC2012 dates at this period is around 1500 years.

**Figure 2. Total deposition and $\Delta^{33}S$ data for the 14 Toba candidate peaks. 2a) shows total deposition for each peak. 2b) Filled bars show the maximum positive and negative fractionations measured. Patterned bars indicate the maximum magnitude $\Delta^{33}S_{volc}$ determined by isotopic mass balance for each peak with samples with $f_{volc} \geq 0.65$, one data point from T3 in EDML with $f_{volc} = 0.63$ has also been included. 2σ errors have been included for $\Delta^{33}S_{meas}$ data and propagated 1σ errors for $\Delta^{33}S_{volc}$.**


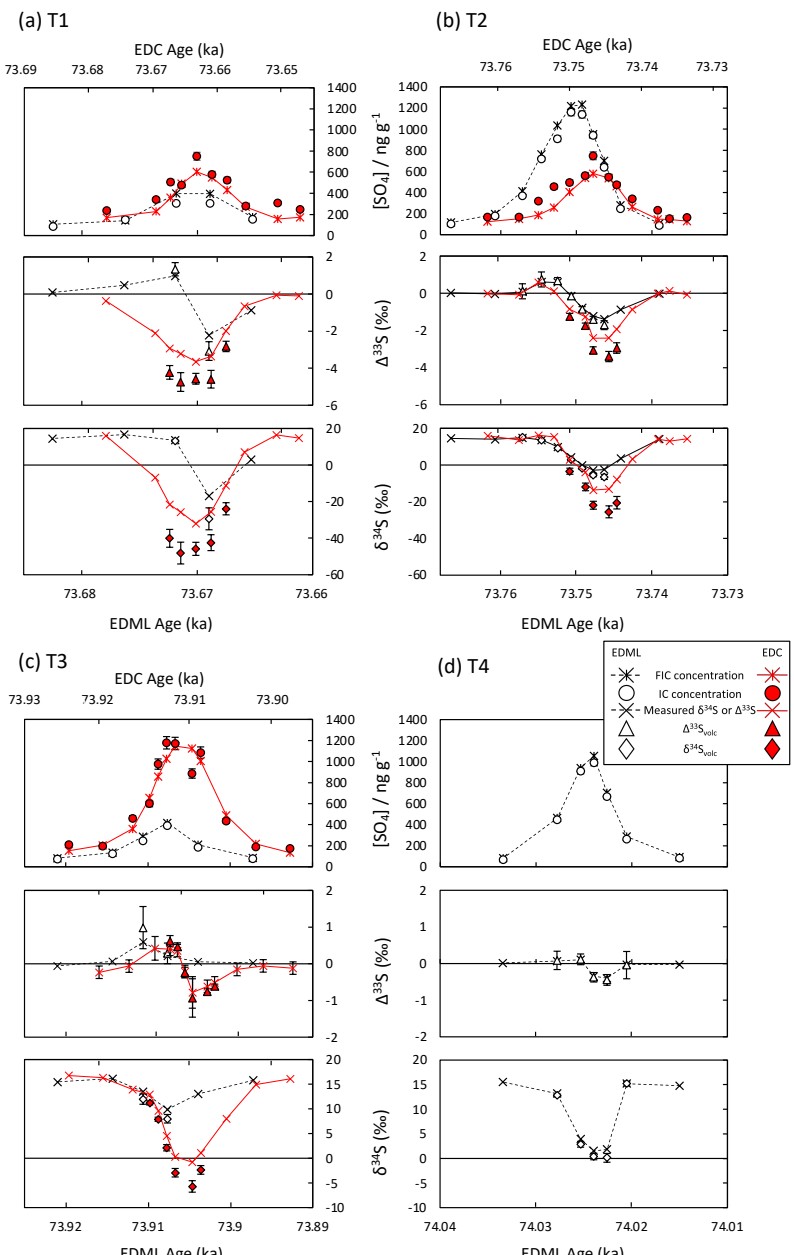

**Figure 3. Sulfur isotopes in sulfate from T1, T2, T3 and T4 in both EDC (a–c) and EDML (all). Red lines and filled symbols are data from the EDC ice core and black represents EDML. Both FIC (lines) and IC (points) data have been plotted for concentration. Lines on the isotope plots show measured values and symbols are background corrected data where $f_{volc} \geq 0.65$, as well as one point from T3 in EDML with $f_{volc} = 0.63$. Error bars indicate $2\sigma$ for measured ratios and $1\sigma$ for background corrected ratios, where error bars are not visible the error is smaller than the symbol. Peaks have been aligned visually based on the isotope records, and this results in minor (<15 y) differences in the AICC2012 age model alignment between the two cores. Note the different scale for the isotopic data in c and d compared to a and b.**

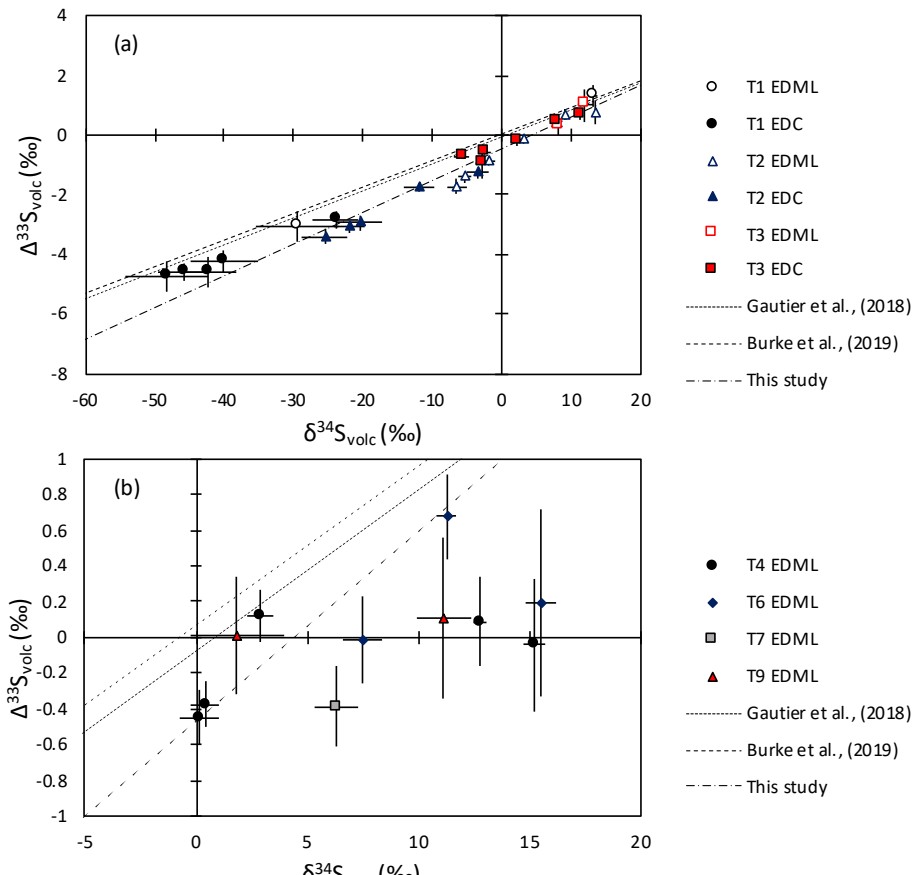

**Figure 4. $\Delta^{33}S_{volc}$ and $\delta^{34}S_{volc}$ for Toba candidates in EDC and EDML. a) Candidates that show a positive correlation indicative of purely stratospheric sulfate, also plotted are linear York regression (York et al., 2004) of the low-latitude stratospheric data (T1–T3) and the regressions from Gautier et al., (2018) and known tropical eruptions from Burke et al., (2019). b) Extratropical candidates with no correlation.**

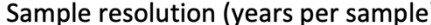

**Figure 5. Comparison between $\Delta^{33}S_{volc}$ measured for volcanic events in this study and the largest volcanic S-MIF from previous studies of historical eruptions. Where available we have also included the sample resolution. Data sources: 1) Savarino et al., (2003), 2) Baroni et al., (2007), 3) Cole-Dai et al., (2009), 4) Gautier et al., (2018), 5) Gautier et al., (2019), 6) Burke et al., (2019), 7) This study. †These dates are taken from Sigl et al., (2015).**


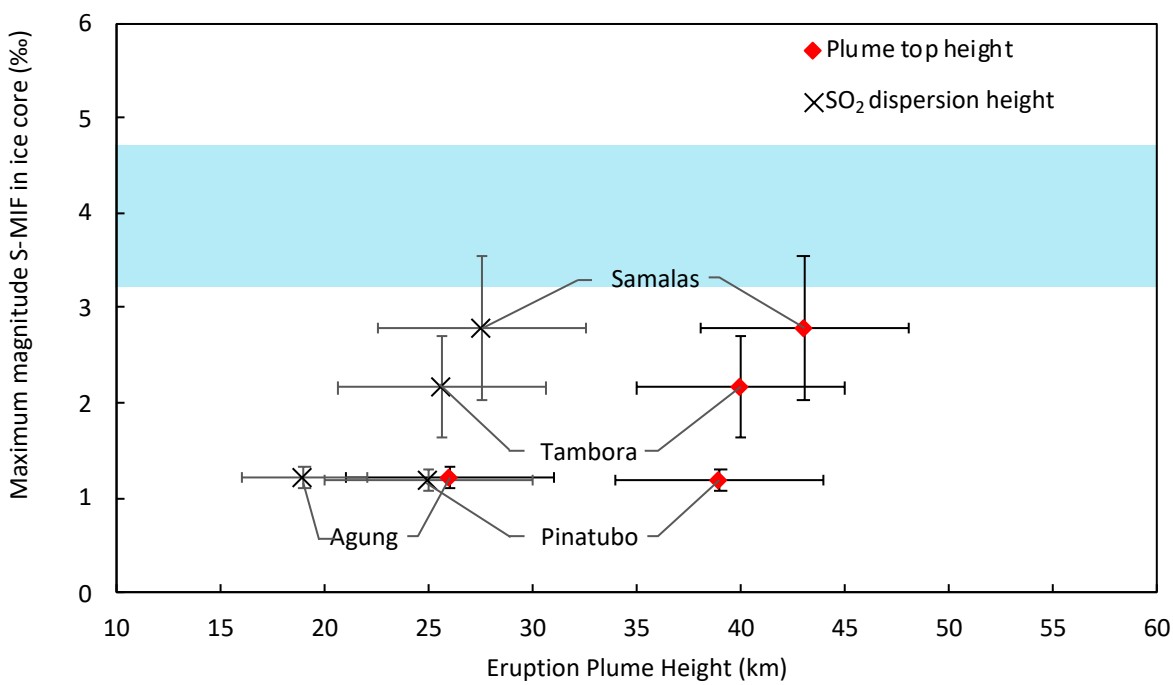

**Figure 6. Maximum S-MIF signal ($\Delta^{33}S$) versus both top plume height and $SO_2$ dispersion height for Common Era events. The highlighted box indicates the region of T1 and T2 MIF signals from this study. Height estimates are from Self and Rampino, (2012) (Agung), Guo et al., (2004) and Holasek et al., (1996) (Pinatubo), Kandlbauer and Sparks, (2014) (Tambora) and Vidal et al., (2016) (Samalas). Where available we have used values for $SO_2$ dispersion height from the literature, otherwise we have used the ratio of $SO_2$ dispersion height to plume top of the Pinatubo 1991 CE eruption as a multiplier, a value of 0.64. $\Delta^{33}S$ data are from Baroni et al., (2007) (Pinatubo, Agung) and Burke et al., (2019) (Tambora, Samalas).**


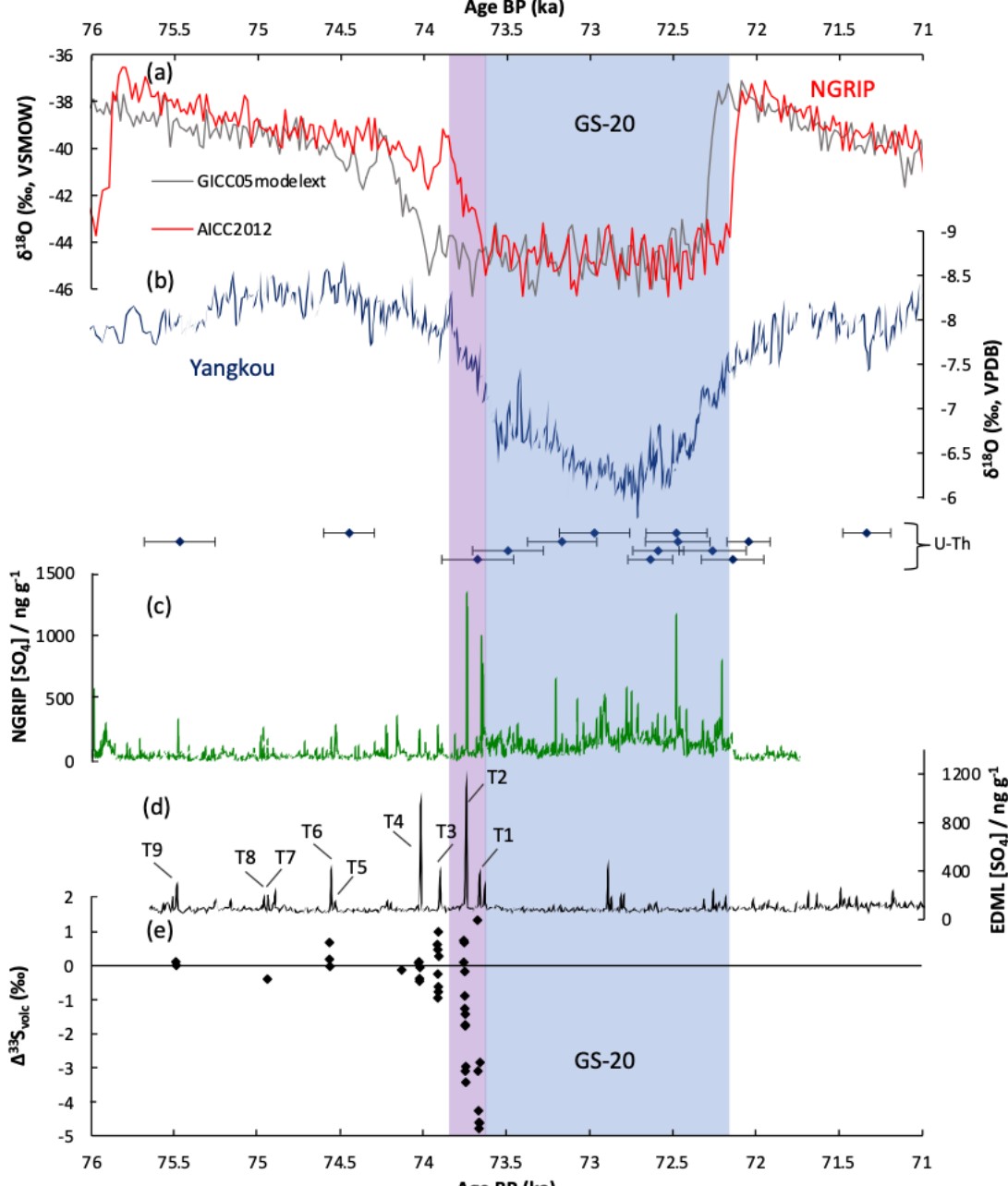

**Figure 7. Paleoclimate indicators across GS-20 (highlighted in blue based upon NGRIP δ¹⁸O with the transition in purple). All ice core data is plotted using the AICC2012 age model unless indicated otherwise. a) Ice core oxygen isotope data from NGRIP (North Greenland Ice Core Project Members, 2004) on both the AICC2012 age model (red) and GICC05modelext age model (Wolff et al., 2010b) (grey) ; b) δ¹⁸O speleothem records from Yangkou, China; the U-Th age data for Yangkou has also been included (blue diamonds) along with their associated 2σ error (Du et al., 2019); c) continuous sulfate concentration in NGRIP and EDML (d) (Svensson et al., 2013); e) Δ³³S$_{volc}$ measured for Toba candidates in this study.**