# Peer review of "New insights into the ~74 ka Toba eruption from sulfur isotopes of polar ice cores"

_Climate of the Past, 2021_

## Referee Comment (RC2)

WRITTEN REPORT

The report by Crick et al. presents new information, in the form of sulfur isotope composition (33S and 34S) of volcanic eruption-sourced sulfate, on a number of volcanic events dated at about 74,000 years BP in two Antarctica ice cores. The sulfur isotope data show unambiguously that most of these events are explosive eruptions in the low latitudes that injected substantial amounts of sulfurous gases into the stratosphere (above the ozone layer). This is based on the findings in the early 2000s (e.g., Savarino et al. 2003; Baroni et al. 2007) that sulfate formed in the stratosphere from oxidation of certain sulfur species (mainly SO2) possesses nonzero sulfur mass-independent fractionation (S-MIF) signatures. The experimental procedures used in this study, including ice core sampling (multiple samples in an event), sulfur isotope ratio measurement, and correction of isotope contribution from non-volcanic sulfate background in the calculation of S-MIF in ice core samples containing both background and volcanic sulfate, follow previously tested and verified methodology and, therefore, the data appear to be robust and of high quality.

From the point of view of identifying large, low latitude stratospheric eruptions in this time period, this study provides additional evidence, but reaches nearly the same conclusion as that by Svenssen et al. (2013), who examined the same group of volcanic events in Antarctica and Greenland cores. Svenssen et al. concluded, based on the simultaneous appearance of large sulfate signals in bipolar ice core records, that nine events (T1-T9) are potential candidates for the famed Toba eruption, dated by Ar isotope geochronometer to be around 73 ka. In this study, the S-MIF data verify that all, maybe with the exception of one, of these events are stratospheric. In addition, a careful examination of the isotope data indicates that locations of events T5-T9 may be extratropical, rather than low latitude.

**Identification of the Toba eruption in ice cores**

The above summary (by me) indicates that this study confirms results from previous work and adds significant new information. However, the main objective of this study (Lines 111-113) appears to be to identify, or to narrow the range of, the signal of the Toba eruption in ice cores. This intent is also suggested by the title of the paper ("the 74 ka Toba eruption"). Unfortunately, sulfur isotope signatures, similar to contemporaneous bipolar sulfate events, do not provide undisputable evidence of a specific eruption, even for Toba. Unlike tephra matching, unambiguous S-MIF data are not a "smoking gun".

In this study, the identification of events T1, T2 and/or T3 as resulting from the Toba eruption is based on two pieces of evidence: the precise timing and the stratospheric nature of the events. Would there be other eruptions that meet these criteria? Or, in other words, can we eliminate the possibility that other eruptions left the volcanic sulfate of these events? The fact that at least three events (T1, T2 and T3) meet these criteria, with the small possibility that they were all left by the same Toba eruption, suggests we cannot be highly confident that the answer is yes. In fact, there are reasons to suspect that none of the candidate events is Toba.

The Toba eruption ejected a huge amount of materials – 3,800 $km^3$ DRE (Costa et al., 2014). This is more than three orders of magnitude that of the 1815 CE Tambora eruption (~ 1.2 $km^3$ DRE, Self et al., 2004). Estimates of the sulfur (aerosol) output of Toba are also several orders of magnitude larger than that of Tambora. (However, I would discount the aerosol

estimates from petrological/volcanological data or ice core data, as these rely on scaling factors (multipliers) that are poorly constrained.) One would expect that the Toba sulfate signal would be exponentially larger than that of Tambora in the same ice core. The volcanic sulfate flux/deposition of all of the three potential Toba events (Figure 2 and Table S1), except for T2 in EDML, is not overwhelmingly large: they are approximately 1-to-2 times that of Tambora. For T2, the much smaller flux (46.2 mg per square m) for EDC suggests that the flux (424) in EDML (about 9 times that of Tambora) may be an outlier. The sulfate flux data of these events in Greenland cores (Svensenn et al., 2013) are also approximately 1-to-2 times that of Tambora. If the Toba eruption resulted in one of the three events, why is its sulfate flux so much smaller than what would be expected? Toba would have to be an exceptionally sulfur-poor eruption to leave one of the three volcanic sulfate signals in ice cores.

In a recently published Holocene volcanic record (Cole-Dai et al., 2021) from the West Antarctica WDC core, ten eruptions in the Holocene are found in the range of 1-2 times of Tambora flux. This suggests that events similar to T1, T2 and T3 in magnitude are not uncommon – they occur relatively frequently, at about one per millennium. The magnitude of ice core signals of T1, T2 and T3 suggests that the eruptions they represent are not at the Toba scale.

The much-smaller-than-expected sulfate signal could be the enigma for identifying Toba in ice cores in ice cores.

**Estimating eruption plume altitude**

The authors of this discussion paper use the extreme cap-delta-33-S values of T1 and T2 to infer that the plume altitude of the eruption clouds must be exceptional high. In fact, they estimate the plume altitude to be at least 45 km for T1 and T2 (Lines 329-330). The estimate is derived or extrapolated from an empirical quantitative relationship between cap-delta-33-S and plume altitude (Figure 6). I question the validity of the extrapolation for two reasons. First, the quantitative relationship is based on four eruptions (Agung, Pinatubo, Samalas and Tambora) or data points with very large uncertainties. The maximum magnitude of cap-delta-33-S for a volcanic event depends strongly on the sampling resolution during the event, as the value of cap-delta-33-S evolves from positive to negative. This is analogous to peak height measurement dependent on sampling resolution during the peak. As a result, I suspect that the uncertainties for maximum cap-delta-33-S values are larger than seen in Figure 6.

Second, the authors cite the study of Lin et al. (2018) to support their proposal that the magnitude of cap-delta-33-S in volcanic sulfate is dependent on the altitude in the stratosphere where the sulfate is formed. I read the Lin et al. paper and have a different understanding of the conclusion regarding that relationship between cap-delta-33-S and altitude. First, Lin et al. measure S-MIF in tropospheric sulfate; therefore, the relationship they describe is for tropospheric altitudes. I think it is quite a stretch to argue that such a relationship could be extrapolated into the stratosphere. Second, Lin et al. explained that the relationship is the result of downward transport of stratospheric sulfate with non-zero cap-delta-33-S; this transport from the stratosphere is supported by an altitude-dependent trend of 35S which is only produced in the stratosphere or above. I think it is on a very shaky ground to use the altitude-S-MIF relationship found by Lin et al. to justify a similar relationship for volcanic sulfate in the stratosphere and to estimate the plume altitude of the volcanic eruption. In my view, interpretation of the volcanic S-MIF magnitude is premature; much more research is required to understand the significance of the volcanic S-MIF magnitude.

**Recommendation**

I would recommend that the paper be revised to (1) de-escalate the certainty that the fallout of the Toba eruption is among the volcanic signals examined, in a fashion similar to what Svenssen et al. did in reaching conclusions regarding Toba identification, and (2) reconsider including estimating eruption plume altitude from the S-MIF data.

---

## Author Response (AR1)

20th August 2021

Dear Editor,

We are grateful for your time in considering our manuscript "New insights into the ~74 ka Toba eruption from sulfur isotopes of polar ice cores". We have amended the text and figures to reflect the thorough and valuable comments from two reviewers and we have responded to all queries raised by them as part of the review process which has greatly improved the manuscript.

We hope that this research will contribute to both the growing understanding of the Toba eruption and the use of sulfur mass-independent fractionation to study volcanic eruptions back through time.

Yours Sincerely,

Laura Crick

**Response to reviewer 1**

*General comment:*

*The study presents the sulfur isotopes of the potential 74 ka Toba eruption sulfate spikes in the Antarctic EDML and EDC ice cores (would be good to mention ice cores in abstract) adding new and valuable information to the ongoing Toba saga. The study is carefully carried out, figures are good and illustrative, the study is well referenced and the message is clear. Nice work, I have only a few minor comments.*

We thank the reviewer for their invaluable insight, and we are grateful for the time spent to consider our manuscript in detail.

Reply to specific comments:

*Specific comments:*

1. *124 onwards: What is the approx. time resolution of the obtained samples? I guess this info can be extracted from the supplementary info, but it would be good to mention in the main text as well.*

   Sampling resolution varies across the individual peaks depending on their sulfate concentration. In general background samples represent 4–8 years of time whereas samples across the highest concentration regions of the sulfate peaks are 1–2 years. Due to diffusion these peaks have broaden and will cover more time than the initial deposition event. This clarification is included in the revised text at L128–131.

2. *154: '…, this integration also corrects for thinning.' It is not entirely clear to me how the integration, that I assume refers to the sulfate peak area, implies a thinning correction? Doesn't this correction need to be done separately after the peak integration? In any case, it would be good to know which thinning models you are applying for the thinning correction (with some reference), and also it would be helpful to know the magnitude of the thinning correction for each core, as this could be quite significant at least for EDML?*

   The following text has been added to the Methods section for clarification (L165–168): "... the total deposition is calculated by integrating over the flux from a given peak. The flux is calculated as the product of the concentration in a slice of ice and the snow accumulation rate. However, as the input data is by depths rather than ages, we multiply by the reciprocal of the annual layer thickness at the depth of the slice. As this annual layer thickness is derived from the age model (Veres et al., 2013), the flux is corrected for thinning during the calculation."

3. *380: You may also compare to the results of (Corrick et al., 2020) for absolute ages.*

   We have amended the text include comparison to Corrick et al., (2020) (L475–480):

"Corrick et al., (2020) provide a comprehensive global compilation of 63 published speleothem records, providing dates of the interstadial transitions of 71,594 ± 230 years for GI-19.2 and 75,583 ± 248 years for GI-20c. In comparison, using the AICC2012 age model we place these transitions at 72,142 years and 75,876 years with the GICC05modelext giving values of 72,340 years and 76,440 years (Rasmussen et al., 2014). Further interrogation of these global records to determine the onset of GS-20 may further improve our estimates for the ages of T1 and T2."

4. *400: 'This would remove…' -> 'This would suggest Toba to be unlikely as a trigger of ….' or similar.*

This alteration is included in the revised text (L504–505).

5. *402: Which candidate gives 3 times the Salamas 1257 CE stratospheric sulfur loading?*

Using the sulfate concentration data for the Antarctic B32 core, we have recalculated the estimates for sulfur loading due to the Toba candidates (L309–322). We estimate a sulfur loading from T2 of 233 Tg S, almost 4 times greater than that of Samalas reported by Toohey and Sigl, (2017) of 59.4 Tg S (L508–509).

*Figure 5: The repeated measurements eg for Salamas have different isotopic amplitudes and are probably obtained for different sample sizes? Would it be possible to show the temporal sample resolution (and maybe the sampled ice core) in the same Figure? In principle the 'true' amplitude of the sulfur isotopes could be extrapolated to infinitesimal sample size? There will still be diffusion in the ice that cannot easiy be accounted for, of course.*

We have amended Figure 5 to reflect the number of samples for each eruption, the ice core used in each study is detailed in the supplementary Table S2. Indeed, the magnitude of S-MIF measured will depend upon the sampling resolution, to investigate this further we have averaged the sample data from Burke et al., (2019) to represent a reduction in sample resolution, the results of which we present in supplementary Figure S8. We use Figure 5 to demonstrate the various studies utilising S-MIF over the past two decades and how our data for volcanic sulfate significantly further back through time compares to Common Era events.

*Figure 7: Strictly speaking the Buizert et al, 2015, publication has nothing to do with the release of the NGRIP isotope profile. A better reference may be (North Greenland Ice Core Project members, 2004).*

Thank you for highlighting this discrepancy, we have amended the references in the main text.

**Response to reviewer 2**

*The report by Crick et al. presents new information, in the form of sulfur isotope composition (33S and 34S) of volcanic eruption-sourced sulfate, on a number of volcanic events dated at about 74,000 years BP in two Antarctica ice cores. The sulfur isotope data show unambiguously that most of these events are explosive eruptions in the low latitudes that injected substantial amounts of sulfurous gases into the stratosphere (above the ozone layer). This is based on the findings in the early 2000s (e.g., Savarino et al. 2003; Baroni et al. 2007) that sulfate formed in the stratosphere from oxidation of certain sulfur species (mainly SO2) possesses nonzero sulfur mass-independent fractionation (S-MIF) signatures. The experimental procedures used in this study, including ice core sampling (multiple samples in an event), sulfur isotope ratio measurement, and correction of isotope contribution from non-volcanic sulfate background in the calculation of S-MIF in ice core samples containing both background and volcanic sulfate, follow previously tested and verified methodology and, therefore, the data appear to be robust and of high quality.*

*From the point of view of identifying large, low latitude stratospheric eruptions in this time period, this study provides additional evidence, but reaches nearly the same conclusion as that by Svenssen et al. (2013), who examined the same group of volcanic events in Antarctica and Greenland cores. Svenssen et al. concluded, based on the simultaneous appearance of large sulfate signals in bipolar ice core records, that nine events (T1-T9) are potential candidates for the famed Toba eruption, dated by Ar isotope geochronometer to be around 73 ka. In this study, the S-MIF data verify that all, maybe with the exception of one, of these events are stratospheric. In addition, a careful examination of the isotope data indicates that locations of events T5-T9 may be extratropical, rather than low latitude.*

We are very grateful the reviewer for the taking the time to thoroughly analyse our article and we will amend the revised text to reflect this feedback. Detailed responses to reviewer comments are addressed below.

**Identification of the Toba eruption in ice cores**

*The above summary (by me) indicates that this study confirms results from previous work and adds significant new information. However, the main objective of this study (Lines 111-113) appears to be to identify, or to narrow the range of, the signal of the Toba eruption in ice cores. This intent is also suggested by the title of the paper ("the 74 ka Toba eruption"). Unfortunately, sulfur isotope signatures, similar to contemporaneous bipolar sulfate events, do not provide undisputable evidence of a specific eruption, even for Toba. Unlike tephra matching, unambiguous S-MIF data are not a "smoking gun".*

While we agree that we cannot provide a smoking gun for Toba without tephra, we can use sulfur isotopes to rule out candidate eruptions based on a muted or non-existent MIF signal. We think that systematic analysis of the candidate sulfate peaks T1–9 provides new and valuable information in the identification of the Toba eruption in the ice core record and provides new insights into its timing and relationship to other paleoclimate records.

*In this study, the identification of events T1, T2 and/or T3 as resulting from the Toba eruption is based on two pieces of evidence: the precise timing and the stratospheric nature of the events. Would there be other eruptions that meet these criteria? Or, in other words, can we eliminate the possibility that other eruptions left the volcanic sulfate of these events? The fact that at least three events (T1, T2 and T3) meet these criteria, with the small possibility that they were all left by the same Toba eruption, suggests we cannot be highly confident that the answer is yes. In fact, there are reasons to suspect that none of the candidate events is Toba.*

We have added further discussion regarding other eruptions which may have resulted in the T1–9 sulfate peaks in section 4.1, lines 285–301:

"The combination of an incomplete geological record of past volcanism along with large uncertainties in dating of geological samples mean that it is not possible to unambiguously attribute a volcanic event with a sulfate deposition event in ice cores in the geological past unless there is a tephra confirmation of the source. We take the approach that given the age estimates of Toba, we can investigate all possible candidates within the age uncertainty and rule out candidate eruptions if they have a muted or weak MIF signal. Although we cannot rule out the possibility that other eruptions deposited these sulfate peaks, providing that the dates of the YTT are accurate, and that it emitted substantial sulfur, then at least one of the candidates we investigated is very likely to be Toba. Using the VOGRIPA database (Crosweller et al., 2012; Brown et al., 2014) we have identified other volcanic events over the age range of T1–9 (when considered on the AICC2012 age model). These volcanic events and their associated dates are detailed in the supplementary Table S4. There are 9 events with VEI ≥ 6 at around 74 ka in VOGRIPA, however they often have large age uncertainties associated with the eruption dates (over 10 ka). Thus, there are many more peaks in addition to T1–T9 in the ice core record that could have been deposited by these eruptions. One of the few with a smaller error is a VEI 6 eruption from the Coatepeque Caldera dated to 72 ± 2 ka (Rose et al., 1999). However, the Toba eruption is the largest of the candidate eruptions over the age range encompassed by T1–T9, and in order to find an eruption with significantly larger S deposition than those considered here at EDC, one would have to extend the search to 79.5 ka, which is well outside the uncertainty in the age of the Toba eruption. Therefore, unless the YTT age and its uncertainty is not accurate or it had negligible sulfur emission, both highly unlikely, then the YTT must have resulted in at least one of the T1–T9 candidates."

*The Toba eruption ejected a huge amount of materials – 3,800 km3 DRE (Costa et al., 2014). This is more than three orders of magnitude that of the 1815 CE Tambora eruption (~ 1.2 km3 DRE, Self et al., 2004). Estimates of the sulfur (aerosol) output of Toba are also several orders of magnitude larger than that of Tambora. (However, I would discount the aerosol estimates from petrological/volcanological data or ice core data, as these rely on scaling factors (multipliers) that are poorly constrained.) One would expect that the Toba sulfate signal would be exponentially larger than that of Tambora in the same ice core. The volcanic sulfate flux/deposition of all of the three potential Toba events (Figure 2 and Table S1), except for T2 in EDML, is not overwhelmingly large: they are approximately 1-to-2 times that of Tambora.*

The cited volume for Tambora is incorrect by a factor of 40 and referencing is not up to date. From Kandlbauer and Sparks, (2014), the Tambora erupted volume is 41 ± 4 km$^3$ DRE and so roughly two (not three) orders of magnitude different to Toba. We are unclear the basis of some of the statements made here. We would not expect Toba sulfate signal to be exponentially larger and are not aware of literature estimating the sulfur output of Toba as several orders of magnitude greater than Tambora; indeed this does not make petrological or geochemical sense given nature of the respective magmas.

With the addition of sulfate data from the B32 Antarctic ice core we have recalculated our sulfur loading estimates to account for the deposition to both EDML and EDC cores (L309–322). We have now included further discussion regarding the potential explanations for a lower sulfur yield for the Toba eruption, see below and lines 332–371 in the revised text:

[revised manuscript text omitted]

An area of future study would be to look at S isotopes in Toba degassed matrix and melt inclusions to reconstruct S degassing of this magma body (Taylor, 1986).

*For T2, the much smaller flux (46.2 mg per square m) for EDC suggests that the flux (424) in EDML (about 9 times that of Tambora) may be an outlier.*

This could be the result of a preservation issue leading to a disparity between the two sites due to the low accumulation rate at EDC site. This issue of preservation has been shown before, for example some cores from the EDC site do not record the 1815 CE Tambora event (Gautier et al., 2016). This clarification has been included in the revised text at lines 180–184.

*The sulfate flux data of these events in Greenland cores (Svensenn et al., 2013) are also approximately 1-to-2 times that of Tambora. If the Toba eruption resulted in one of the three events, why is its sulfate flux so much smaller than what would be expected? Toba would*

*have to be an exceptionally sulfur-poor eruption to leave one of the three volcanic sulfate signals in ice cores.*

As detailed above, some eruptions - such as the Millennium Eruption – despite being large magnitude can deposit comparatively little sulfate to the ice core. In addition, if the Toba eruption was comprised of multiple eruptions, the bulk rock estimates would suggest a single large event because of the temporal resolution of geological ages, while the ice cores could record multiple sulfur peaks. Totalling the sulfur loading estimated from the ice cores for T1, T2 and T3 returns estimates of nearly 8 times that of the sulfur loading due to Samalas (459 Tg S vs 59.4 Tg S) and over 16 times greater than Tambora (28.1 Tg S). This is further clarified at lines 443–446.

*The much-smaller-than-expected sulfate signal could be the enigma for identifying Toba in ice cores in ice cores.*

Indeed, we agree this is a difficulty if the sulfate deposition to the ice cores due to Toba is small in relative to its estimated magnitude. However, the resulting sulfate peaks would still need to be within the dating estimate and bipolar, leaving T1, T2 and T3 as the best candidates for the Toba eruption in the ice cores.

**Estimating eruption plume altitude**

*The authors of this discussion paper use the extreme cap-delta-33-S values of T1 and T2 to infer that the plume altitude of the eruption clouds must be exceptional high. In fact, they estimate the plume altitude to be at least 45 km for T1 and T2 (Lines 329-330). The estimate is derived or extrapolated from an empirical quantitative relationship between cap-delta-33-S and plume altitude (Figure 6). I question the validity of the extrapolation for two reasons. First, the quantitative relationship is based on four eruptions (Agung, Pinatubo, Samalas and Tambora) or data points with very large uncertainties. The maximum magnitude of cap-delta-33-S for a volcanic event depends strongly on the sampling resolution during the event, as the value of cap-delta-33-S evolves from positive to negative. This is analogous to peak height measurement dependent on sampling resolution during the peak. As a result, I suspect that the uncertainties for maximum cap-delta-33-S values are larger than seen in Figure 6.*

Alongside alterations to Figure 5 as recommended by the first reviewer to show the sample resolution for each study cited there, we clarify in the revised text the effect of sampling resolution upon the magnitude of S-MIF signal measured (L416–423). As such for Figure 6 we have reported S-MIF values for studies with high sampling resolution so we are more likely to record close to the maximum $\Delta^{33}S$ for a given eruption (Agung 1.13 years/sample, Pinatubo 0.7 years/sample, Tambora 0.16 years/sample and Samalas 0.23 years/sample). As an additional test we averaged values from the highest resolution eruptions (Tambora and Samalas) to mimic a reduction in sample resolution. From this test the magnitude of the $\Delta^{33}S$ signal decreased slightly, we have included a figure in the supplement (Fig. S8) which demonstrates this test. Following helpful discussions with Thomas Aubry (see below), we have also amended Figure 6 to show the distinction between the $SO_2$ dispersion height and the plume top height, with additional discussion (L408–414).

*Second, the authors cite the study of Lin et al. (2018) to support their proposal that the magnitude of cap-delta-33-S in volcanic sulfate is dependent on the altitude in the stratosphere where the sulfate is formed. I read the Lin et al. paper and have a different understanding of the conclusion regarding that relationship between cap-delta-33-S and altitude. First, Lin et al. measure S-MIF in tropospheric sulfate; therefore, the relationship they describe is for tropospheric altitudes. I think it is quite a stretch to argue that such a relationship could be extrapolated into the stratosphere.*

We interpret the Lin et al., (2018) study to conclude that due to the presence of cosmogenic $^{35}$S in the samples, that they are measuring sulfur that derived from the stratosphere, even though they collected it in the troposphere. As written in Lin et al. (2018) *"We do not rule out the possibility that there is an unknown SO2 oxidation mechanism which mass-independently enriches $^{33}$S in sulfate products in the free troposphere, but, at present, there is no evidence for the existence of such a process. Consequently, we favor the explanation by which downward transport of stratospheric sulfates is the most plausible source of positive $\Delta^{33}$S values in tropospheric sulfates (11–15)"*.

*Second, Lin et al. explained that the relationship is the result of downward transport of stratospheric sulfate with non-zero cap-delta-33-S; this transport from the stratosphere is supported by an altitude-dependent trend of 35S which is only produced in the stratosphere or above. I think it is on a very shaky ground to use the altitude-S-MIF relationship found by Lin et al. to justify a similar relationship for volcanic sulfate in the stratosphere and to estimate the plume altitude of the volcanic eruption. In my view, interpretation of the volcanic S-MIF magnitude is premature; much more research is required to understand the significance of the volcanic S-MIF magnitude.*

We consider that the simplest explanation for the Lin data is an altitude dependent $\Delta^{33}$S in the modern atmosphere. As written in Lin et al. (2018) *"The altitude-dependent variation of $\Delta^{33}$S revealed by enrichment of stratospherically sourced $^{35}$S indicates that sulfate aerosols originating from the higher atmosphere possess a greater $\Delta^{33}$S value than the boundary layer"*.

Although we agree that much more research is needed, the data from the eruptions we do have measurements from supports our interpretation. We have refrained from fitting a line to the data, since there are large uncertainties, but we use it to highlight future potential research avenues for S-MIF studies. We have specifically highlighted these large uncertainties with this interpretation in the text below. Further analysis, such as $\Delta^{17}$O measurements, would be required to make more robust conclusions regarding the plume height achieved by the Toba eruption. We agree that this interpretation is still in its speculative stage. Thus, we have edited the corresponding section of text to read as follows:

"When we compare the S-MIF signals for our Toba candidates, particularly T1, T2, and T3, to previous studies of Common Era events we find that the larger magnitude events, such as Samalas and Tambora, have larger magnitude MIF signals (Fig. 5). Independent geological estimates of eruption plume height are available for numerous Common Era eruptions (Aubry et al., 2021), determined by a range of methods including using cloud positions from satellite sensors for the SO₂ injection altitude (Guo et al., 2004) and modelling lithic clast dispersal for the plume top height (Sigurdsson and Carey, 1989). There is a positive

correlation between the maximum $\Delta^{33}S$ value measured and plume top height for Common Era eruptions that have high resolution (≥5 samples/peak) S isotope measurements from ice cores (Burke et al., 2019) or snow pits (Baroni et al., 2007) (Fig. 6). If this relationship is extrapolated to the range of MIF signals measured for T1 and T2, we would anticipate the plume top height to be in excess of 45 km, similar to the model estimate of the Toba eruption of ~ 42 km from Costa et al., (2014). A more appropriate eruptive parameter to compare with the maximum $\Delta^{33}S$ is the $SO_2$ dispersion height, as this is the altitude of the sulfur plume and thus is likely be the altitude at which the S-MIF is inherited. However the $SO_2$ dispersion height is not as well constrained for past eruptions as the plume top height (Aubry et al., 2021). For the Agung and Pinatubo eruptions, we have used the $SO_2$ dispersion altitudes from the literature, compiled in the IVESPA database (Aubry et al., 2021; ivespa.co.uk). For the Tambora and Samalas eruptions we have calculated the $SO_2$ dispersion height using the ratio of $SO_2$ height to plume top height for the Pinatubo eruption (0.64). With this data we place the $SO_2$ dispersion height for Toba at over 30 km (Fig.6).

Although this tentative relationship between $\Delta^{33}S$ and plume height supports the conclusion of an altitude dependence of S-MIF by Lin et al., (2018), there are some important caveats to note. For instance, the maximum magnitude of the $\Delta^{33}S$ measured in the ice will depend on the sampling resolution and preservation in the core. We have tested the impact of decreasing sampling resolution by averaging samples from Tambora and Samalas from Burke et al. (2019). In these two instances, the lower sample resolution reduces the $\Delta^{33}S_{volc}$ signal slightly as expected, but the average remains within error of the measured maximum S-MIF value (see Fig. S8). This exercise further illustrates the importance of maximizing sampling resolution, which is made possible by measurement with MC-ICP-MS (Burke et al., 2019)."

Furthermore, our estimates of plume height or $SO_2$ dispersion height are based on a tentative relationship from only a handful of Common Era eruptions with large uncertainties associated with whether the S-MIF measurements captured the maximum magnitude of the S-MIF signal (see Supplementary Info, Fig. S8). Thus this relationship should be further investigated and validated. One such method that could provide additional information on plume height is sulfate oxygen MIF measurements ($\Delta^{17}O$), since O-MIF in OH in the stratosphere varies with altitude (Zahn et al., 2006) which can then be inherited by sulfate during oxidation (Gautier et al., 2019)." (L428–430).

**Recommendation**
*I would recommend that the paper be revised to (1) de-escalate the certainty that the fallout of the Toba eruption is among the volcanic signals examined, in a fashion similar to what Svenssen et al. did in reaching conclusions regarding Toba identification, and (2) reconsider including estimating eruption plume altitude from the S-MIF data.*

Thank you for your thorough analysis and recommendations. To address point 1), if none of the sulfate signals measured were from Toba then either 1) the sulfur loading from Toba is even lower than we have suggested here or 2) the Ar/Ar dates are inaccurate and/or their uncertainties are underestimated. Since there are no other large bipolar events in the ice cores within the age uncertainty of the YTT date, one would have to extend the search to

79.5 ka, which is well outside the reported uncertainty in the age of the Toba eruption. Our comparison to radiometrically dated speleothem records shows that the absolute age of the AICC2012 timescale is within the uncertainty of the U-series ages (~200 years). Thus, if the Ar/Ar dates and uncertainties are accurate then at least one of these candidates must be associated with the Toba eruption. Given that other large eruptions such as the Millennium Eruption also have relatively low sulfur yields, we think that a low sulfur yield for Toba is a more likely explanation than inaccurate Ar/Ar ages. Regarding the estimate of plume altitude, as noted above we have expanded further upon the caveats associated with this estimate and reiterate the need for further research regarding the relationship between S-MIF and eruption plume altitude. Following discussions with Thomas Aubry we have modified Figure 6 to display the plume top height and have included further data with the $SO_2$ dispersion height for each Common Era eruption, an important distinction for climate modelling.

Finally, we have included additional analysis of the B32 Antarctic core which, due to its proximity to the EDML ice core, has allowed us to further compare the Toba candidates preserved in EDML to Common Era eruptions in B32 and recalculate the sulfur loading due to Toba (L175–176; L309–322).

**References**

[revised manuscript text omitted]